# POISSON-DIRAC NEURAL NETWORKS FOR MODELING COUPLED DYNAMICAL SYSTEMS ACROSS DOMAINS

**Razmik Arman Khosrovian**[*1] **Takaharu Yaguchi**[2] **Hiroaki Yoshimura**[3] **Takashi Matsubara**[*4]
[1]Osaka University  [2]Kobe University  [3]Waseda University  [4]Hokkaido University

## ABSTRACT

Deep learning has achieved great success in modeling dynamical systems, providing data-driven simulators to predict complex phenomena, even without known governing equations. However, existing models have two major limitations: their narrow focus on mechanical systems and their tendency to treat systems as monolithic. These limitations reduce their applicability to dynamical systems in other domains, such as electrical and hydraulic systems, and to coupled systems. To address these limitations, we propose Poisson-Dirac Neural Networks (PoDiNNs), a novel framework based on the Dirac structure that unifies the port-Hamiltonian and Poisson formulations from geometric mechanics. This framework enables a unified representation of various dynamical systems across multiple domains as well as their interactions and degeneracies arising from couplings. Our experiments demonstrate that PoDiNNs offer improved accuracy and interpretability in modeling unknown coupled dynamical systems from data.

## 1 INTRODUCTION

Deep learning has achieved great success in modeling dynamical systems (Chen et al., 2018; Anandkumar et al., 2020), following its successes in image processing and natural language processing (He et al., 2016; Vaswani et al., 2017). It provides a data-driven approach for predicting the behavior of complex dynamical systems, even when governing equations are unknown (Kasim & Lim, 2022; Matsubara & Yaguchi, 2023). These models function as computational simulators and show promise in applications across diverse fields, including weather forecasting, mechanical design, and system control (Lam et al., 2023; Pfaff et al., 2020; Horie et al., 2021).

However, Greydanus et al. (2019) identified a key limitation of data-driven models: they accumulate modeling errors in long-term predictions, leading to rapid failure. Hamiltonian Neural Networks (HNNs) were proposed to overcome this limitation by incorporating Hamiltonian mechanics. Inspired by this approach, many studies have developed neural network models that not only learn the superficial dynamics from data, but also adhere to fundamental laws of physics. Examples include Lagrangian Neural Networks (LNNs) (Cranmer et al., 2020), Neural Symplectic Forms (NSFs) (Chen et al., 2021), Poisson Neural Networks (PNNs) (Jin et al., 2022),Constrained HNNs (CHNNs) (Finzi et al., 2020), and Dissipative SymODENs (Zhong et al., 2020), as shown in Table 1.

Despite recent progress, two key limitations remain in modeling dynamical systems, especially those described by ordinary differential equations (ODEs). The first limitation is the narrow focus on mechanical systems. Models applied to other domains, such as electric circuits or magnetic fields, often fail to leverage the governing principles of those systems, such as Kirchhoff's current and voltage laws (Jin et al., 2022; Matsubara & Yaguchi, 2023). The second limitation is that most methods treat the system as a single, monolithic entity. In reality, many systems consist of interacting components, such as robot arms with multiple joints or electric circuits with various elements (Yoshimura & Marsden, 2006a). Although the port-Hamiltonian formulation theoretically addresses these interactions, no prior work has fully leveraged its potential. These limitations prevent current methods from effectively handling multiphysics scenarios, where systems from different domains interact, such as those involving DC motors (van der Schaft & Jeltsema, 2014; Gay-Balmaz & Yoshimura, 2023).

---

[*]Correspondence to `u113287d@ecs.osaka-u.ac.jp` and `matsubara@ist.hokudai.ac.jp`.

Table 1: Comparison of Methods for Modeling Dynamical Systems

| Model | Formulation | Identifying Coupling Patterns | Applicable to Multiphysics | Degeneracy | Dissipation | External Inputs | Coordinate-Free |
|---|---|---|---|---|---|---|---|
| HNN (Greydanus et al., 2019) | canonical Hamiltonian | ✗ | ✗ | ✗ | ✗ | ✗ | ✗ |
| LNN (Cranmer et al., 2020) | Lagrangian | ✗ | ✗ | ✗ | ✗ | ✗ | ✗ |
| NSF (Chen et al., 2021) | general Hamiltonian | ✗ | ✗ | ✗ | ✗ | ✗ | ✓ |
| CHNN (Finzi et al., 2020) | constrained canonical Hamiltonian | ✗ | ✗ | ✳ | ✗ | ✗ | ✗ |
| CLNN (Finzi et al., 2020) | constrained Lagrangian | ✗ | ✗ | ✳ | ✗ | ✗ | ✗ |
| PNN (Jin et al., 2022) | Poisson | ✗ | ✗ | ✓ | ✗ | ✗ | ✓ |
| Dis. SymODEN (Zhong et al., 2020) | port-Hamiltonian on Darboux coordinates | ✗ | ✗ | ✗ | ✓ | † | ✗ |
| PoDiNN (proposed) | Poisson-Dirac with ports | ✓ | ✓ | ✓ | ✓ | ✓ | ✓ |

✳Available only for known holonomic constraints.    † Available only for external force on mass.

To address these limitations, we propose *Poisson-Dirac Neural Networks (PoDiNNs)*, which leverage the Dirac structure to unify the port-Hamiltonian and Poisson formulations (Courant, 1990; Duindam et al., 2009; van der Schaft, 1998; van der Schaft & Jeltsema, 2014; Yoshimura & Marsden, 2006a). The Dirac structure explicitly represents the coupling between internal and external components. A con-

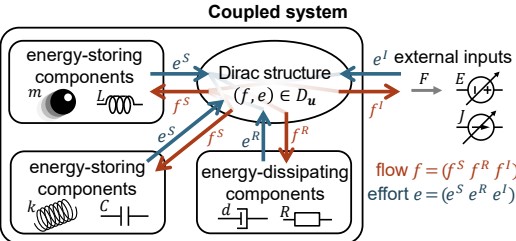

Figure 1: Conceptual diagram of PoDiNNs.

ceptual diagram is shown in Fig. 1. The advantages of PoDiNNs are summarized below and in Table 1, and are validated through experiments on seven simulation datasets spanning mechanical, rotational, electro-magnetic, and hydraulic domains.

**Identifying Coupling Patterns**   Many real-world systems are coupled systems, consisting of interacting components (Yoshimura & Marsden, 2006a). While existing methods represent these as single vector fields or energy functions, PoDiNNs explicitly learn internal couplings as bivectors on vector bundles, separate from component-wise characteristics modeled by neural networks. This approach improves both interpretability and modeling accuracy.

**Applicable to Various Domains of Systems**   The Dirac structure allows PoDiNNs to describe various dynamical systems across multiple domains. While most neural network models of ODEs focus on mechanical systems, PoDiNNs can model interactions of mechanical, rotational, electrical, and hydraulic systems, addressing real-world multiphysics scenarios.

**Unifying Various Aspects of Dynamical Systems**   Previous models have addressed specific aspects of dynamical systems, such as degenerate dynamics (Finzi et al., 2020; Jin et al., 2022), energy dissipation, external inputs (Zhong et al., 2020), and coordinate transformations (Chen et al., 2021; Jin et al., 2022). PoDiNNs offer a unified framework to address all these aspects together by leveraging the Dirac structure.

## 2   BACKGROUND THEORY AND RELATED WORK

Attempts to learn ODEs have a long history (Nelles, 2001); see also Appendix A for a broader review. Neural Ordinary Differential Equations (Neural ODEs) transformed this field (Chen et al., 2018), which train networks to approximate the vector field (the right-hand side of the ODE), which is then integrated numerically. However, they are often overly general and fail to capture dynamical systems governed by principles such as energy conservation (Greydanus et al., 2019). To address this, many refinements have been proposed by integrating insights from analytical mechanics.

### 2.1   HAMILTONIAN SYSTEMS

**Canonical Hamiltonian Systems**   The Hamiltonian formulation of mechanics defines Hamilton's equations of motion, using a pair of $n$-dimensional generalized coordinates $\boldsymbol{q} = (q_1, \ldots, q_n)$ and

momenta $\boldsymbol{p} = (p_1, \dots, p_n)$ as the state. Given an energy function $H$, the system evolves as:

$$\dot{q}_i = \frac{\partial H}{\partial p_i}, \quad \dot{p}_i = -\frac{\partial H}{\partial q_i}, \text{ or equivalently, } \begin{pmatrix} \dot{\boldsymbol{q}} \\ \dot{\boldsymbol{p}} \end{pmatrix} = \begin{pmatrix} O & I \\ -I & O \end{pmatrix} \nabla H(\boldsymbol{q}, \boldsymbol{p}), \quad (1)$$

where $\dot{}$ denotes the time derivative, $I$ and $O$ are the $n$-dimensional identity and zero matrices, respectively, and $\nabla H$ denotes the gradient of $H$. Systems describable by Hamilton's equations are Hamiltonian systems. See Appendix B.1 for an example.

HNNs were introduced to learn such systems, with a neural network $H_{NN}$ serving as the energy function $H$ (Greydanus et al., 2019). HNNs have shown greater long-term robustness than Neural ODEs (Chen et al., 2022; Gruver et al., 2022).

**Hamiltonian Systems of Coordinate-Free Form** From the differential geometry perspective, the equations of motion in Eq. (1) are defined on the cotangent bundle $\mathcal{M} = T^*\mathcal{Q}$ of the configuration space $\mathcal{Q} = \mathbb{R}^n$ with the standard symplectic form $\Omega = \sum_i \mathrm{d}q_i \wedge \mathrm{d}p_i$, where $\boldsymbol{q} \in \mathcal{Q}$ and $\boldsymbol{p} \in T^*_{\boldsymbol{q}}\mathcal{Q}$, respectively. Here, $T^*_{\boldsymbol{q}}\mathcal{Q}$ denotes the cotangent space to $\mathcal{Q}$ at $\boldsymbol{q}$, and $\wedge$ denotes the exterior product. The symplectic form $\Omega$ leads to a skew-symmetric linear bundle map $\Omega^\flat_{\boldsymbol{u}} : T_{\boldsymbol{u}}\mathcal{M} \to T^*_{\boldsymbol{u}}\mathcal{M}$ at each point $\boldsymbol{u} \in \mathcal{M}$, which satisfies $\langle \Omega^\flat_{\boldsymbol{u}}(\boldsymbol{v}), \boldsymbol{w} \rangle = \Omega_{\boldsymbol{u}}(\boldsymbol{v}, \boldsymbol{w})$ for any $\boldsymbol{u} \in \mathcal{M}$ and $\boldsymbol{v}, \boldsymbol{w} \in T_{\boldsymbol{u}}\mathcal{M}$, where $\langle \cdot, \cdot \rangle$ denotes the natural pairing. We will denote by the subscript $_{\boldsymbol{u}}$ an assignment at point $\boldsymbol{u}$ and omit the subscripts if an equation holds independently of the point $\boldsymbol{u}$. Given a smooth function $H : \mathcal{M} \to \mathbb{R}$, its differential is denoted by $\mathrm{d}H$. A vector field $X$ on $\mathcal{M}$ assigns a tangent vector $X_{\boldsymbol{u}} \in T_{\boldsymbol{u}}\mathcal{M}$ to each point $\boldsymbol{u} \in \mathcal{M}$. Then, Hamilton's equations in the coordinate-free form defines a vector field $X_H$ called the Hamiltonian vector field by

$$\Omega^\flat(X_H) = \mathrm{d}H. \quad (2)$$

$X_H$ defines the time evolution of the state $\boldsymbol{u}$ as $\dot{\boldsymbol{u}}(t) = (X_H)_{\boldsymbol{u}(t)}$ at time $t$, and $\boldsymbol{u}(t)$ over a certain period is a solution of the system. Then, the tuple $(H, \Omega, X_H)$ is called a Hamiltonian system. A Hamiltonian system with the standard symplectic form $\Omega$ is said to be canonical, and its coordinate system is known as Darboux coordinates.

Hamilton's equations also describe the same dynamics using generalized velocities $\boldsymbol{v} \in T_{\boldsymbol{q}}\mathcal{Q}$ instead of generalized momenta $\boldsymbol{p} \in T^*_{\boldsymbol{q}}\mathcal{Q}$, i.e., on the tangent bundle $T\mathcal{Q}$ rather than the cotangent bundle $T^*\mathcal{Q}$. In this case, the symplectic form $\Omega$ is replaced by a Lagrangian 2-form (Marsden & Ratiu, 1999). From another viewpoint, the symplectic form $\Omega$ defines the coordinate system. Lagrangian Neural Networks (LNNs) learn the dynamics on the tangent bundle $T\mathcal{Q}$ (Cranmer et al., 2020). Neural Symplectic Forms (NSFs) learn the symplectic form directly from data, generalizing HNNs and LNNs for arbitrary coordinate systems (Chen et al., 2021). In general, Hamiltonian systems preserve the energy $H$, as $\mathcal{L}_{X_H} H = \langle \mathrm{d}H, X_H \rangle = \Omega(X_H, X_H) = 0$, where $\mathcal{L}_{X_H}$ denotes the Lie derivative along $X_H$, and the last equality follows from the skew-symmetry of $\Omega$.

**Poisson Systems** Because the symplectic form $\Omega$ is non-degenerate in the sense that the bundle map $\Omega^\flat_{\boldsymbol{u}}$ is non-degenerate, it leads to a 2-tensor $B$ called a Poisson bivector satisfying $B(\mathrm{d}H, \mathrm{d}G) = \Omega(X_H, X_G)$ for any smooth functions $H, G$ on $\mathcal{M}$. $B$ leads to a skew-symmetric linear bundle map $B^\sharp_{\boldsymbol{u}} : T^*_{\boldsymbol{u}}\mathcal{M} \to T_{\boldsymbol{u}}\mathcal{M}$ at each point $\boldsymbol{u} \in \mathcal{M}$, which satisfies $B_{\boldsymbol{u}}(\boldsymbol{\alpha}, \boldsymbol{\beta}) = \langle \boldsymbol{\alpha}, B^\sharp_{\boldsymbol{u}}(\boldsymbol{\beta}) \rangle$ for any $\boldsymbol{u} \in \mathcal{M}$ and $\boldsymbol{\alpha}, \boldsymbol{\beta} \in T^*_{\boldsymbol{u}}\mathcal{M}$. Then, it holds that $B^\sharp = (\Omega^\flat)^{-1}$. Hamilton's equations, Eq. (2), are rewritten as

$$X_H = B^\sharp(\mathrm{d}H). \quad (3)$$

The tuple $(H, B, X_H)$ is called a Poisson system. On the Darboux coordinates, $B = \sum_i \frac{\partial}{\partial p_i} \wedge \frac{\partial}{\partial q_i}$. Lie-Poisson neural networks were proposed to learn this system with known $B$ (Eldred et al., 2024).

## 2.2 Degeneracy, Dissipation, and External Inputs

**Degenerate Systems** If the state $\boldsymbol{u}$ is constrained to a submanifold $\tilde{\mathcal{M}} \subset \mathcal{M}$, Hamilton's equations do not directly describe the dynamics. For example, consider a pair of mass-spring systems, indexed by $i \in \{1, 2\}$, with a constraint $q_1 = q_2$ such that the two masses are coupled and always have the same displacement and velocity. While the Hamiltonian is the sum of those of coupled systems, Hamilton's equations $\Omega^\flat(X_H) = \mathrm{d}H$ with the standard symplectic form $\Omega$ do not describe the dynamics. There are two primary methods for handling such degenerate Hamiltonian systems.

The first approach uses coordinate transformations to reduce the system's degrees of freedom and define a submanifold $\tilde{\mathcal{M}} \subset \mathcal{M}$, where Hamilton's equations describe the dynamics using the standard symplectic form $\Omega$ or Poisson bivector $B$. The Darboux-Lie theorem ensures the local existence of such transformations. The standard Poisson bivector $B$ on $\tilde{\mathcal{M}}$ can also be expressed on $\mathcal{M}$ while its bundle map is degenerate, and it represents the coordinate transformation. Hence, degenerate Hamiltonian systems are included into Poisson systems. See Appendix B.2 for an example. Jin et al. (2022) introduced Poisson neural networks (PNNs) that combine neural network-based coordinate transformations with SympNets for degenerate systems (Dinh et al., 2017; Jin et al., 2020). However, this approach lacks interpretability due to non-unique, nonlinear transformations, and extending it to systems with external inputs or dissipation is challenging.

The other approach introduces constraint forces from the coupling, defining constrained Hamiltonian systems. Finzi et al. (2020) proposed constrained HNNs (CHNNs) for such systems. However, this method is limited to holonomic constraints (e.g., constraints on configurations). Furthermore, as the constraints are predetermined, it remains unclear how to learn these constraints directly from data.

**Port-Hamiltonian Systems**    To incorporate dissipation and external inputs, some studies have explored the port-Hamiltonian formulation with neural networks (Zhong et al., 2020). Although some have mentioned the underlying Dirac structure (Neary & Topcu, 2023; Di Persio et al., 2024), they rely on the following canonical form without fully leveraging its flexibility:

$$\begin{pmatrix} \dot{\boldsymbol{q}} \\ \dot{\boldsymbol{p}} \end{pmatrix} = \left( \begin{pmatrix} O & I \\ -I & O \end{pmatrix} - \begin{pmatrix} O & O \\ O & D(\boldsymbol{q}) \end{pmatrix} \right) \nabla H(\boldsymbol{q}, \boldsymbol{p}) + \begin{pmatrix} \boldsymbol{0} \\ G(\boldsymbol{q}) \end{pmatrix} \boldsymbol{f}, \tag{4}$$

where $D \in \mathbb{R}^{n \times n}$ represents dissipation, $\boldsymbol{f} \in \mathbb{R}^m$ is the control input vector, and $G \in \mathbb{R}^{n \times m}$ is the control input matrix. This formulation has several disadvantages. The matrix $\begin{pmatrix} O & I \\ -I & O \end{pmatrix}$ represents the standard symplectic form $\Omega = \sum_i \mathrm{d}q_i \wedge \mathrm{d}p_i$ on $\mathbb{R}^{2n}$, which cannot handle degeneracies. Dissipation from multiple sources, such as dampers or friction, is condensed into a single term $D$, limiting interpretability and modeling performance. Furthermore, this formulation cannot handle externally defined velocities, as seen in models of buildings shaken by the ground. While Eidnes et al. (2023) employed a non-standard form to address the first limitation, their method still inherits the latter two.

## 3    POISSON-DIRAC NEURAL NETWORKS

To overcome the limitations of existing methods, we propose Poisson-Dirac Neural Networks (PoDiNNs), which leverage the Dirac structure to unify Poisson and port-Hamiltonian systems. Detailed background theory and proofs of the following theorems are provided in Appendix D.

### 3.1    DIRAC STRUCTURE

Let $V$ be an $n$-dimensional vector space and $V^*$ its dual space, with the natural pairing $\langle \cdot, \cdot \rangle$ between them. Define the symmetric pairing $\langle\!\langle \cdot, \cdot \rangle\!\rangle$ on $V \oplus V^*$ as

$$\langle\!\langle (\boldsymbol{v}, \boldsymbol{\alpha}), (\bar{\boldsymbol{v}}, \bar{\boldsymbol{\alpha}}) \rangle\!\rangle = \langle \boldsymbol{\alpha}, \bar{\boldsymbol{v}} \rangle + \langle \bar{\boldsymbol{\alpha}}, \boldsymbol{v} \rangle \quad \text{for} \quad (\boldsymbol{v}, \boldsymbol{\alpha}), (\bar{\boldsymbol{v}}, \bar{\boldsymbol{\alpha}}) \in V \oplus V^*$$

where $\oplus$ denotes the direct sum of two vector spaces (Courant, 1990).

**Definition 1** (Courant (1990); Yoshimura & Marsden (2006a)). *A Dirac structure on a vector space $V$ is a vector subspace $D \subset V \oplus V^*$ such that $D = D^\perp$, where $D^\perp$ is the orthogonal complement of $D$ with respect to the pairing $\langle\!\langle \cdot, \cdot \rangle\!\rangle$.*

Since $D = D^\perp$, $\langle \boldsymbol{\alpha}, \bar{\boldsymbol{v}} \rangle + \langle \bar{\boldsymbol{\alpha}}, \boldsymbol{v} \rangle = 0$ for any $(\boldsymbol{v}, \boldsymbol{\alpha}), (\bar{\boldsymbol{v}}, \bar{\boldsymbol{\alpha}}) \in D$, and hence $\langle \boldsymbol{\alpha}, \boldsymbol{v} \rangle = 0$ for any $(\boldsymbol{v}, \boldsymbol{\alpha}) \in D$. A typical Dirac structure can be constructed as follows:

**Theorem 1** (Courant (1990); Yoshimura & Marsden (2006a)). *Let $V$ be a vector space and $\Delta$ a vector subspace. Define the annihilator $\Delta^\circ$ of $\Delta$ as $\Delta^\circ = \{ \boldsymbol{\alpha} \in V^* \mid \langle \boldsymbol{\alpha}, \boldsymbol{v} \rangle = 0 \text{ for all } \boldsymbol{v} \in \Delta \} \subset V^*$. Then, $D_V = \Delta \oplus \Delta^\circ \subset V \oplus V^*$ is a Dirac structure on $V$.*

A vector bundle $\mathcal{F}$ over a manifold $\mathcal{M}$ is a collection of vector spaces $\mathcal{F}_{\boldsymbol{u}}$, called fibers, smoothly assigned to points $\boldsymbol{u} \in \mathcal{M}$, where the manifold $\mathcal{M}$ is called the base space. A typical example is the tangent bundle $T\mathcal{M}$, where the fiber at $\boldsymbol{u}$ is the tangent space $T_{\boldsymbol{u}}\mathcal{M}$, and its dual bundle is the cotangent bundle $T^*\mathcal{M}$. The Whitney sum $\oplus$ defines a vector bundle whose fiber at each point is the direct sum of the fibers of the two bundles at that point. Given a vector bundle $\mathcal{F}$ over $\mathcal{M}$ and its dual $\mathcal{E} = \mathcal{F}^*$, the Dirac structure is defined as a subbundle of $\mathcal{F} \oplus \mathcal{E}$.

Table 2: Categorization of Components in Different Domains

| Domain | Mechanical | | Rotational | | Electro-Magnetic | | Hydraulic |
|---|---|---|---|---|---|---|---|
| Subdomain | Potential | Kinetic | Potential | Kinetic | Electric | Magnetic | Potential |
| flow (input) | velocity | force | angular velocity | torque | current | voltage | volume flow rate |
| effort (output) | force | velocity | torque | angular velocity | voltage | current | pressure |
| state | displacement | momentum | angle | angular momentum | electric charge | magnetic flux | volume |
| energy-storing | spring | mass | (potential) | inertia | capacitor | inductor | hydraulic tank |
| energy-dissipating | damper | – | friction | – | resistor | resistor | — |
| external input | external force | moving boundary | external torque | – | voltage source | current source | incoming fluid flow |

**Definition 2** (Courant (1990); Yoshimura & Marsden (2006a); van der Schaft & Jeltsema (2014)).
*Consider a vector bundle $\mathcal{F}$ over a manifold $\mathcal{M}$. A distribution $\Delta$ is a collection of vector subspaces $\Delta_{\boldsymbol{u}} \subset \mathcal{F}_{\boldsymbol{u}}$, each assigned smoothly to $\mathcal{M}$ at point $\boldsymbol{u}$, forming a vector subbundle of $\mathcal{F}$. The annihilator $\Delta^\circ$ of $\Delta$ is a collection of the annihilators $\Delta_{\boldsymbol{u}}^\circ$ of $\Delta_{\boldsymbol{u}}$, also forming a subbundle of $\mathcal{E} = \mathcal{F}^*$. Then, a Dirac structure $D$ is constructed as $D = \Delta \oplus \Delta^\circ$, which is a subbundle of $\mathcal{F} \oplus \mathcal{E}$.*

If $\mathcal{F} \oplus \mathcal{E} = TM \oplus T^*\mathcal{M}$, the Dirac structure $D$ can reformulate Hamiltonian, Poisson, and constrained Hamiltonian systems (see Appendix D). Here, we assume that the fibers $\mathcal{F}_{\boldsymbol{u}}$ and $\mathcal{E}_{\boldsymbol{u}}$ of the vector bundles $\mathcal{F}$ and $\mathcal{E}$ at $\boldsymbol{u}$ are decomposed as

$$\mathcal{F}_{\boldsymbol{u}} = \mathcal{F}_{\boldsymbol{u}}^S \oplus \mathcal{F}_{\boldsymbol{u}}^R \oplus \mathcal{F}_{\boldsymbol{u}}^I \quad \text{and} \quad \mathcal{E}_{\boldsymbol{u}} = \mathcal{E}_{\boldsymbol{u}}^S \oplus \mathcal{E}_{\boldsymbol{u}}^R \oplus \mathcal{E}_{\boldsymbol{u}}^I, \tag{5}$$

where $\mathcal{F}_{\boldsymbol{u}}^S = T_{\boldsymbol{u}}\mathcal{M}$ and $\mathcal{E}_{\boldsymbol{u}}^S = T_{\boldsymbol{u}}^*\mathcal{M}$. A point on the fibers $\mathcal{F}_{\boldsymbol{u}}$ and $\mathcal{E}_{\boldsymbol{u}}$ is denoted by $\boldsymbol{f} = (\boldsymbol{f}^S, \boldsymbol{f}^R, \boldsymbol{f}^I)$ and $\boldsymbol{e} = (\boldsymbol{e}^S, \boldsymbol{e}^R, \boldsymbol{e}^I)$, respectively. We refer to $\boldsymbol{f} \in \mathcal{F}_{\boldsymbol{u}}$ as *flows*, $\boldsymbol{e} \in \mathcal{E}_{\boldsymbol{u}}$ as *efforts*, and both collectively as *port variables*. Note that our definitions mainly followed those in Duindam et al. (2009); van der Schaft & Jeltsema (2014), while we can find other definitions in Yoshimura & Marsden (2006a).

**Theorem 2.** *Consider vector bundles $\mathcal{F}$ and $\mathcal{E} = \mathcal{F}^*$ over $\mathcal{M}$. The collection of*

$$D_{\boldsymbol{u}} = \{(\boldsymbol{f}, \boldsymbol{e}) \in \mathcal{F}_{\boldsymbol{u}} \times \mathcal{E}_{\boldsymbol{u}} \mid \boldsymbol{f} = B_{\boldsymbol{u}}^\sharp(\boldsymbol{e})\}$$

*for the bundle map $B_{\boldsymbol{u}}^\sharp : \mathcal{E}_{\boldsymbol{u}} \to \mathcal{F}_{\boldsymbol{u}}$ of a bivector $B$ is a Dirac structure $D \subset \mathcal{F} \oplus \mathcal{E}$.*

Here, we define PoDiNNs as a special case of the Poisson-Dirac formulation (Courant, 1990).

**Definition 3** (Poisson-Dirac Neural Network). *Let $\mathcal{F} \oplus \mathcal{E}$ be a vector bundle over a manifold $\mathcal{M}$ defined in Eq. (5), which assigns to each $\boldsymbol{u} \in \mathcal{M}$ a vector space $(\mathcal{F}_{\boldsymbol{u}}^S \oplus \mathcal{F}_{\boldsymbol{u}}^R \oplus \mathcal{F}_{\boldsymbol{u}}^I) \times (\mathcal{E}_{\boldsymbol{u}}^S \oplus \mathcal{E}_{\boldsymbol{u}}^R \oplus \mathcal{E}_{\boldsymbol{u}}^I)$. Let $D \subset \mathcal{F} \oplus \mathcal{E}$ be a Dirac structure defined in Theorem 2. Let $H : \mathcal{M} \to \mathbb{R}$ be an energy function, which determines the effort $\boldsymbol{e}^S = \mathrm{d}H \in \mathcal{E}_{\boldsymbol{u}}^S$. The effort $\boldsymbol{e}^R \in \mathcal{E}_{\boldsymbol{u}}^R$ is determined by a mapping $R_{\boldsymbol{u}} : \mathcal{F}_{\boldsymbol{u}}^R \to \mathcal{E}_{\boldsymbol{u}}^R$ of the flow $\boldsymbol{f}^R \in \mathcal{F}_{\boldsymbol{u}}^R$. The effort $\boldsymbol{e}^I(t) \in \mathcal{E}_{\boldsymbol{u}}^I$ is a time-dependent function. The functions $H$ and $R_{\boldsymbol{u}}$ are implemented using neural networks. If for each $\boldsymbol{u}(t)$ and $t \in [a, b]$, it holds that*

$$((\boldsymbol{f}^S(t), \boldsymbol{f}^R(t), \boldsymbol{f}^I(t)), (\boldsymbol{e}^S(t), \boldsymbol{e}^R(t), \boldsymbol{e}^I(t))) \in D_{\boldsymbol{u}(t)},$$

*the tuple $(H, B, R, \boldsymbol{e}^I, \boldsymbol{f}^S)$ is called Poisson-Dirac Neural Networks (PoDiNNs).*

## 3.2 FLOWS AND EFFORTS FOR COMPONENTS

The point $\boldsymbol{u}$ at the base space $\mathcal{M}$ represents the states of the dynamics, which include the displacement of a spring, the momentum of a mass, the angle and angular momentum of a rotating rod, the electric charge of a capacitor. Intuitively, flows $\boldsymbol{f}$ are the inputs to components, while efforts $\boldsymbol{e}$ are the outputs from the components. Components considered in our formulation are summarized in Table 2, with concrete examples in Appendices C and F.

Similar to Hamiltonian and Poisson systems, the flow $\boldsymbol{f}^S \in T_{\boldsymbol{u}}\mathcal{M}$ represents the vector field $X_H$ on $\mathcal{M}$, defining the time evolution $\dot{\boldsymbol{u}}$ of the state $\boldsymbol{u}$. The effort $\boldsymbol{e}^S \in T_{\boldsymbol{u}}^*\mathcal{M}$ corresponds to the differential $\mathrm{d}H$ of the Hamiltonian $H : \mathcal{M} \to \mathbb{R}$. The superscript $^S$ indicates that these components store energy. In electric circuits, capacitors and inductors are examples, with states as electric charge and magnetic flux, and efforts as the voltages across and currents through them, respectively. In PoDiNNs, these components are modeled using neural networks that replace the energy functions $H$, similar to HNNs and Dissipative SymODENs.

The flow $\boldsymbol{f}^R \in \mathcal{F}_{\boldsymbol{u}}^R$ and effort $\boldsymbol{e}^R \in \mathcal{E}_{\boldsymbol{u}}^R$ represent energy-dissipating components, such as dampers and resistors. A damper's flow $f^R$ is the velocity (i.e., the rate of extension or compression), and its effort $e^R$ is the force, which are related as $e^R = -df^R$ for a linear damper with a damping coefficient $d$. Unlike a spring, a damper does not store its own energy, so its flow $f^R$ is not on $\mathcal{F}_{\boldsymbol{u}}^S$ but on $\mathcal{F}_{\boldsymbol{u}}^R$. The superscript $^R$ indicates that these components are "resistive." However, components that supply energy, such as special diodes, can also be classified into this category. In any case, each component is implemented in PoDiNNs by a neural network that approximates its characteristic $R_{\boldsymbol{u}} : \boldsymbol{f}^R \mapsto \boldsymbol{e}^R$.

The flow $\boldsymbol{f}^I \in \mathcal{F}_{\boldsymbol{u}}^I$ and effort $\boldsymbol{e}^I \in \mathcal{E}_{\boldsymbol{u}}^I$ represent external inputs, such as an external force (where force is the effort) or a moving boundary (where velocity is the effort). The superscript $^I$ indicates that they are "inputs." Their efforts $\boldsymbol{e}^I$ depend only on time $t$, not on other components. Their flows $\boldsymbol{f}^I$ are not required for determining the system's dynamics but represent the outcomes of external inputs, such as the reaction force exerted on the moving boundary. In PoDiNNs, these external inputs are fed into the neural networks.

### 3.3 BIVECTOR FOR REPRESENTING COUPLED SYSTEMS

For the coordinates $q_i$ and $p_i$ on $\mathcal{M}$, the basis vectors of the tangent space $T_{\boldsymbol{u}}\mathcal{M}$ are $\frac{\partial}{\partial q_i}$ and $\frac{\partial}{\partial p_i}$, respectively. The $i$-th basis vectors of $\mathcal{F}_{\boldsymbol{u}}^R$, $\mathcal{F}_{\boldsymbol{u}}^I$, $\mathcal{E}_{\boldsymbol{u}}^R$, and $\mathcal{E}_{\boldsymbol{u}}^I$ are denoted by $\xi_i^R$, $\xi_i^I$, $\xi_i^{R*}$, and $\xi_i^{I*}$, respectively. The bivector $B$ assigns to each point $\boldsymbol{u} \in \mathcal{M}$ wedge products of the basis vectors of the flow space $\mathcal{F}_{\boldsymbol{u}}$, such as $\frac{\partial}{\partial p_i} \wedge \frac{\partial}{\partial q_j}$, $\frac{\partial}{\partial p_i} \wedge \xi_j^R$, and $\frac{\partial}{\partial p_i} \wedge \xi_j^I$, thereby defining the coupling patterns among components. For example, $\frac{\partial}{\partial p_i} \wedge \xi_j^I$ couples the $j$-th external input with the $i$-th mass $m_i$ with the state $p_i$. The effort of the $j$-th external input is expressed as $e_j^I \xi_j^{I*}$ when the basis is explicit. This is fed to the bivector $\frac{\partial}{\partial p_i} \wedge \xi_j^I = -\xi_j^I \wedge \frac{\partial}{\partial p_i}$, resulting in $-e_j^I \frac{\partial}{\partial p_i}$, which forms part of the flow $f_i^S \frac{\partial}{\partial p_i}$ for mass $m_i$. Therefore, we can make the following remarks.

**Remark 1** (Coupling as Non-zero Elements of Bivector). *Coupling between two components is represented by a non-zero bivector element, which links the effort of one to the flow of the other. Thus, by learning the bivector $B$ from the observations of the target system, we can identify the coupling patterns between the system's components.*

**Remark 2** (Degeneracy of Dynamics as Degeneracy of Bundle Map). *Constraints between components that cause degenerate dynamics are reflected in degeneracy of the bundle map $B^\sharp$. Thus, by learning the bivector $B$ from the observations of the target system and examining how it degenerates, we can identify the constraints imposed on the system.*

**Remark 3** (Coordinate Transformation by Bivector). *The bivector $B$ defines the coordinate system, which allows PoDiNNs to learn system dynamics regardless of the coordinate system used for the observations.*

**Remark 4** (Multiphysics). *Our formulation can represent systems across various domains, as summarized in Table 2.*

These remarks are fundamental in system identification and can aid in reverse engineering, as the coupling patterns of circuit elements serve as representations of the circuit diagrams. Also, PoDiNNs are the first neural-network method to handle multiple domains of dynamical systems and their interactions by leveraging the Dirac structure. See Appendix C for concrete examples.

### 3.4 DISCUSSIONS FOR COMPARISONS AND LIMITATIONS

As discussed above, PoDiNNs are the first model to cover degenerate dynamics, dissipation, external inputs, and coordinate transformations in a unified manner.

Previous models, including HNNs, LNNs, NSFs, PNNs, and Dissipative SymODENs, approximate the Hamiltonian $H$, Lagrangian $L$, or dissipative term $D$ using neural networks, which implicitly learn the relationships between variables. However, these relationships are difficult to extract due to the implicit nature of the learning and the high nonlinearity of the networks. In contrast, PoDiNNs separate the coupling patterns $B$ from the energy functions $H$, improving interpretability and generalization performance.

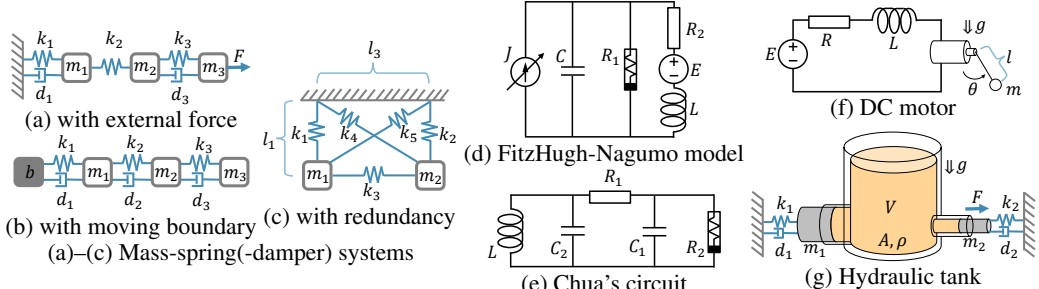

Figure 2: Diagrams of systems that provide datasets. Detailed definitions are found in Appendix F.

In an absolute coordinate system, the displacements of springs were computed from the absolute positions of springs' edges internally within the energy function. Although PoDiNNs can handle such system, the bivector for energy-storing components takes the standard form $B = \sum_i \frac{\partial}{\partial p_i} \wedge \frac{\partial}{\partial q_i}$, which hinders the identification of their coupling patterns. PoDiNNs are still able to identify coupling patterns involving energy-dissipating components and external inputs. To identify the coupling patterns between energy-storing components, it is necessary to employ a relative coordinate system based on displacements, rather than absolute positions.

LNNs and NSFs are designed for non-canonical Hamiltonian systems, where the Hamiltonian vector field $X_H$ is implicitly defined as $\Omega^\flat(X_H) = \mathrm{d}H$ (Cranmer et al., 2020; Chen et al., 2021). These require matrix inversion to compute $X_H$, which is computationally expensive, numerically unstable, and are not directly applicable to degenerate dynamics. In contrast, PoDiNNs explicitly define $X_H$ as part of $\boldsymbol{f} = B^\sharp(\boldsymbol{e})$, offering faster and more stable computations and handling degeneracy.

In mechanical systems, energy-dissipating components such as dampers and friction are characterized by first setting the velocities set, from which the corresponding forces are then derived. As a result, flow is always defined as velocity, and effort as force. In the electric circuits, however, the flow for resistors and diodes can be either current or voltage, depending on their coupling with other components. In practice, it is advisable to include an abundance of both types of components. Any excess components will either be ignored or exhibit redundant characteristics, as demonstrated in the experiments.

The dynamics of electric circuits are generally described by differential-algebraic equations (DAEs). For example, when a capacitor is connected to a direct voltage source in parallel, infinite current instantaneously flows into the capacitor, and its voltage matches that of the direct voltage source. This behavior cannot be represented by ODEs alone. While PoDiNNs cannot fully represent such systems, they can still describe a wide range of systems and expand the scope of modeling unknown dynamical systems from observations.

See also Appendix E for implementation and further discussions.

## 4 EXPERIMENTS AND RESULTS

### 4.1 EXPERIMENTAL SETTINGS

**Datasets** We evaluated PoDiNNs and related methods to demonstrate their modeling performance using seven simulation datasets, as shown in Fig. 2. Due to page limitations, we briefly overview their characteristics. The full explanations can be found in Appendix F.

For the mechanical domain, we prepared three mass-spring(-damper) systems (a)–(c), with the velocities of the masses as observations. In the absolute coordinate system, we used the absolute positions of the springs' ends as the springs' states, and in the relative coordinate system, their displacements. As external inputs, an external force $F$ is applied to system (a), and a moving boundary $b$ is coupled with system (b). Dissipative SymODENs cannot directly account for the latter. System (c) has redundant observations; while springs are coupled only with two masses, the displacements of all five springs were provided as observations. CHNNs cannot model this redundancy, but PNNs can. In all three systems, the springs and dampers exhibit nonlinear characteristics.

We selected two nonlinear electric circuits, (d) the FitzHugh-Nagumo model and (e) Chua's circuit, from the electro-magnetic domain (Izhikevich & FitzHugh, 2006; Chua, 2007). The capacitor voltage and inductor current were used as the observations.

We also used two multiphysics systems (f) and (g). In system (f), a DC motor bridges an electric circuit in the electro-magnetic domain and a pendulum in the rotational domain. In system (g), a hydraulic tank in the hydraulic domain is connected to two cylinders with pistons in the mechanical domain, each of which is also connected to a fixed wall via a spring and damper. An external force is applied to the smaller piston, which moves the larger piston through the fluid in the tank.

**Implementation Details**    We implemented all experimental code from scratch using Python v3.11.9, along with numpy v1.26.4, scipy v1.12.1, pytorch v2.3.1, and desolver v4.1.1 (Paszke et al., 2017). See also Appendix F for more details.

We compared Neural ODEs, Dissipative SymODENs, PNNs, and PoDiNNs. Neural ODEs were evaluated on all datasets. Dissipative SymODENs were tested on systems (a) and (b) in the absolute coordinate system, while PNNs were evaluated on system (c). Other combinations were out of scope of the original studies. For PNNs, we used HNNs in place of SympNets for a fair comparison. For Dissipative SymODENs and PoDiNNs, we assumed that the kinetic energy in the mechanical domain could be expressed as $\frac{1}{2m}p^2$, where $m$ is the mass and $p$ is the momentum. Therefore, we employed this form with a learnable parameter $m$, rather than a neural network. The same approach was applied to capacitors and inductors for PoDiNNs. All other components were assumed to be nonlinear and were modeled using neural networks. In the absolute coordinate system, the potential energy was modeled for all configurations $q_1, \ldots, q_n$ plus the position $q_b$ of the moving boundary collectively by a single neural network. In the relative coordinate system or other domains, the potential energy was modeled separately for each component. We assumed that the number of energy-dissipating components and the nature of their flows (e.g., current or voltage in electric circuits) are known, and also examined the impact of inaccurate assumptions.

Each model was trained using one-step predictions on the training subset. Specifically, given a random state snapshot $\boldsymbol{u}^{(n)}$ at $n$-th step, each model predicted the next state $\tilde{\boldsymbol{u}}^{(n+1)}$ after a time step $\Delta t$. Then, all parameters were updated to minimize the squared error between the predicted state $\tilde{\boldsymbol{u}}^{(n+1)}$ and the ground truth $\boldsymbol{u}^{(n+1)}$, normalized by state standard deviations. It is known that longer prediction steps can improve robustness against noise (Chen et al., 2020). However, as we aimed to purely compare the representational performance of models, no noise was added to the datasets. We confirmed that longer prediction steps only led to performance degradation.

**Evaluation Metrics**    We evaluated models using the accuracy of the solution to the initial value problem on the test subset. Starting from the initial value of each trajectory, each model predicted the entire trajectory of $N$ steps and calculated the mean squared error (MSE) between the predicted state $\tilde{\boldsymbol{u}}^{(n)}$ and the ground truth $\boldsymbol{u}^{(n)}$ at each step indexed by $n$. The mean of these MSEs across all trajectories and all time steps was computed as the evaluation metric, referred to as the overall MSE;

$$MSE(\tilde{\boldsymbol{u}}; \boldsymbol{u}) = \frac{1}{N} \sum_{n=1}^{N} [\mathrm{MSE}(\tilde{\boldsymbol{u}}^{(n)}, \boldsymbol{u}^{(n)})]. \tag{6}$$

Lower values indicate better performance. Additionally, we defined the valid prediction time (VPT) as the ratio of the number of steps taken before the MSE first exceeds a certain threshold $\theta$ to the total length $N$ of the test trajectory (Botev et al., 2021; Jin et al., 2020; Vlachas et al., 2020);

$$VPT(\tilde{\boldsymbol{u}}; \boldsymbol{u}) = \frac{1}{N} \arg\max_{n_f} \{n_f | \mathrm{MSE}(\tilde{\boldsymbol{u}}^{(n)}, \boldsymbol{u}^{(n)}) < \theta \text{ for all } n \leq n_f\}. \tag{7}$$

Higher values indicate better performance. The threshold $\theta$ was set to ensure that most models achieved VPTs between 0.1 and 0.9. For each dataset, models were trained and evaluated from scratch over 10 trials per dataset.

## 4.2    RESULTS

**Numerical Performance**    The results in Table 3 show that PoDiNNs consistently outperformed all other models across all datasets and metrics. VPTs show that PoDiNNs provided stable predictions for durations 1.5 to 20 times longer. These differences were statistically significant, with p-values less than 0.0005, as evaluated by the Mann-Whitney U test. In some cases, the difference in MSE is small, while that in VPT is substantial. This happens when a model ignores oscillations in the

Table 3: Experimental Results.

| | Mass-Spring-Damper Systems | | | | | | | | | |
|---|---|---|---|---|---|---|---|---|---|---|
| **Dataset** | with external force | | | | with moving boundary | | | | (c) with redundancy | |
| **Coordinate** | (a) relative | | (a') absolute | | (b) relative | | (b') absolute | | relative | |
| **Model** | MSE↓ | VPT↑ | MSE↓ | VPT↑ | MSE↓ | VPT↑ | MSE↓ | VPT↑ | MSE↓ | VPT↑ |
| Neural ODE | 4.90±0.27 | 0.128±0.021 | 7.68±1.07 | 0.097±0.008 | 7.43±1.19 | 0.153±0.039 | 5.02±0.56 | 0.135±0.052 | 2490.61±1847.24 | 0.099±0.004 |
| HNN Variant* | — | | 8.31±0.56 | 0.104±0.017 | — | | 5.92±0.12 | 0.001±0.000 | 634.22±300.01 | 0.000±0.000 |
| PoDiNN | **4.33**±0.26 | **0.622**±0.002 | **7.02**±0.49 | **0.437**±0.053 | **0.26**±1.12 | **0.856**±0.015 | **3.74**±0.84 | **0.581**±0.040 | **0.11**±0.02 | **0.863**±0.017 |
| | $\times 10^{-1}$ | $\theta = 10^{-3}$ | $\times 10^{-1}$ | $\theta = 10^{-3}$ | $\times 10^{-2}$ | $\theta = 10^{-4}$ | $\times 10^{-2}$ | $\theta = 10^{-4}$ | $\times 10^{-1}$ | $\theta = 10^{-3}$ |

| | Electric Circuits | | | | Multiphysics | | | |
|---|---|---|---|---|---|---|---|---|
| | (d) FitzHugh-Nagumo | | (e) Chua's | | (f) DC Motor | | (g) Hydraulic Tank | |
| **Model** | MSE↓ | VPT↑ | MSE↓ | VPT↑ | MSE↓ | VPT↑ | MSE↓ | VPT↑ |
| Neural ODE | 48.96±17.43 | 0.322±0.041 | 14.74±1.33 | 0.287±0.016 | 16.03±5.15 | 0.276±0.168 | 30.62±8.22 | 0.045±0.010 |
| PoDiNN | **1.64**±1.37 | **0.649**±0.072 | **9.21**±0.83 | **0.469**±0.010 | **2.11**±2.90 | **0.923**±0.013 | **5.42**±3.65 | **0.918**±0.013 |
| | $\times 10^{-4}$ | $\theta = 10^{-3}$ | $\times 10^{-1}$ | $\theta = 10^{-3}$ | $\times 10^{-3}$ | $\theta = 10^{-4}$ | $\times 10^{-2}$ | $\theta = 10^{-4}$ |

Each score represents the median over 10 trials, followed by the ± symbol and the quartile deviation. *Dissipative SymODENs for systems (a) and (b), and PNNs for system (c).

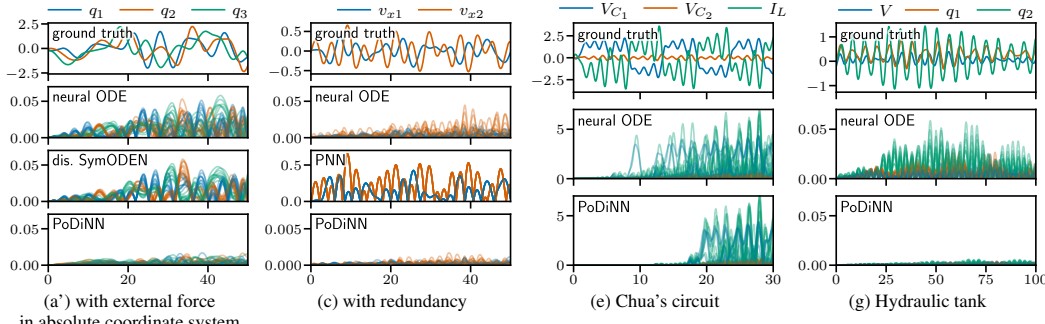

(a') with external force in absolute coordinate system   (c) with redundancy   (e) Chua's circuit   (g) Hydraulic tank

Figure 3: Visualizations of example results. Each top panel shows ground truth trajectories, while the other panels show the absolute errors of all 10 trials in semi-transparent color. See also Fig. A1.

trajectory and outputs an average path, which keeps the MSE low but results in poor VPT. This suggests that VPT is a more reliable metric than MSE (Botev et al., 2021; Jin et al., 2020; Vlachas et al., 2020). We show example trajectories and absolute errors in Figs. 3 and A1.

PoDiNNs perform well both in the absolute and relative coordinate systems, showing their adaptability to different coordinate systems. Even in the absolute coordinate system, PoDiNNs decompose dissipations and external inputs into coupling patterns and individual characteristics, providing a more effective inductive bias, whereas Dissipative SymODENs treat dissipations and external inputs using black-box functions $D$ and $G$.

PoDiNNs demonstrate excellent accuracy in system (c) because they successfully simplify the dynamics by identifying the degeneracy from high-dimensional observations. In contrast, PNNs struggled to learn the appropriate coordinate transformation. Although the Darboux-Lie theorem guarantees the existence of such a transformation, it does not imply that learning it is straightforward. Also, Neural ODEs lack guarantees for energy conservation, resulting in diverging trajectories.

**Identifying Coupling Patterns and Component Characteristics (e)** We examined how the bivector $B$ identifies coupling patterns in system (e), Chua's circuit. When two coefficients differed by a factor of 1000 or more, we considered the larger one as a detected coupling and the smaller one as effectively zero, indicating no coupling. In all 10 trials, we obtained the bivector $B = -a\frac{\partial}{\partial \psi} \wedge \frac{\partial}{\partial Q_2} + b\xi_1^R \wedge \frac{\partial}{\partial Q_2} - c_1\xi_1^R \wedge \frac{\partial}{\partial Q_1} + c_2\xi_2^R \wedge \frac{\partial}{\partial Q_1}$, with trial-wise positive parameters $a$, $b$, $c_1$, and $c_2$. We emphasize that the coefficients of other bivector elements, such as $\frac{\partial}{\partial \psi} \wedge \frac{\partial}{\partial Q_1}$ and $\xi_2^R \wedge \frac{\partial}{\partial Q_2}$, were effectively zero. Because $R_1$ and $R_2$ cannot be directly observed and their indices are interchangeable, they were appropriately reordered for analysis.

We normalized the coefficients of the bivector elements and the characteristics of system components, as their scales cancel each other out. Then, from the learned bivector $B$, we can derive Kirchhoff's current laws, $I_{C_1} = -I_{R_1} + I_{R_2}$ and $I_{C_2} = -I_L + I_{R_1}$, and Kirchhoff's voltage laws, $V_L = V_{C_2}$, $V_{R_1} = V_{C_1} - V_{C_2}$, and $V_{R_2} = -V_{C_1}$. This allows us to construct a circuit diagram, which perfectly matches that of Chua's circuit, shown in Fig. 2 (e).

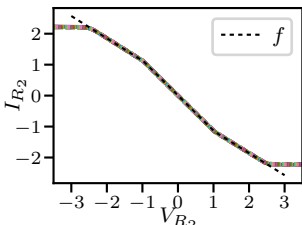

Figure 4: The identified characteristics of $R_2$.

While coupling patterns in more complex circuits may not be unique due to Norton's and Thevenin's theorems, our formulation can learn one valid realization. Figure 4 illustrates the input-output relationship of the neural network representing $R_2$. The results from all 10 trials, shown as colored dashed lines, almost perfectly overlap the true relationship, shown as a black dashed line, within the range of $\pm 2.5$. This demonstrates that PoDiNNs accurately identified the component characteristics, even though its response was never observed directly. The region beyond this range was not included in the training subset, so it is expected that PoDiNNs did not learn the relationship there.

**Identifying Coupling Patterns of System (g)** We also examined the bivector $B$ identified in system (g), hydraulic tank. In all 10 trials, we consistently obtained the bivector $B = \frac{\partial}{\partial p_1} \wedge \frac{\partial}{\partial V} - 0.3(\frac{\partial}{\partial p_2} \wedge \frac{\partial}{\partial V}) + \frac{\partial}{\partial p_1} \wedge \frac{\partial}{\partial q_1} + \frac{\partial}{\partial p_2} \wedge \frac{\partial}{\partial q_2}$ between the energy-storing components, with the coefficients 1.0 or $-0.3$, accurate to five significant figures. The coefficients of the first two terms, 1.0 and $-0.3$, correspond to the cross-sectional areas $a_1$ and $a_2$ of two cylinders attached at the bottom of the tank, with the negative sign indicating that the flow directions are opposite. The remaining two terms represent the couplings between the masses and springs. All other couplings—between the dampers and masses, $\frac{\partial}{\partial p_1} \wedge \frac{\partial}{\partial d_1}$ and $\frac{\partial}{\partial p_2} \wedge \frac{\partial}{\partial d_2}$, and between the external force and the mass, $\frac{\partial}{\partial p_2} \wedge \frac{\partial}{\partial \xi^s}$—were also identified with non-zero coefficients, even though the overall scale is indeterminate due to the cancellation of gravitational acceleration, fluid density, and masses.

**Impact of Number of Hidden Components (b)** The states of energy-storing components are provided as observations, and external inputs are typically known (or inferred using methods like neural CDE (Kidger et al., 2020)). PoDiNNs also require specifying the number and type of energy-dissipating components, which are usually unknown. To assess the impact of the assumed number of components, $n_d$, we tested system (b) in the relative coordinate system (see Table 4). The correct number of dampers is $n_d = 3$. When $n_d < 3$, performance was extremely poor, indicating incorrect dynamics due to missing dampers. When $n_d > 3$, performance was

Table 4: Impact of # Components and VPT.

| PoDiNNs | Training | Test |
|---|---|---|
| $n_d = 0$ | $0.005 \pm 0.000$ | $0.000 \pm 0.000$ |
| $n_d = 1$ | $0.009 \pm 0.002$ | $0.001 \pm 0.000$ |
| $n_d = 2$ | $0.015 \pm 0.000$ | $0.001 \pm 0.000$ |
| $n_d = 3$ | $\mathbf{0.925} \pm 0.007$ | $\mathbf{0.581} \pm 0.040$ |
| $n_d = 4$ | $\mathbf{0.932} \pm 0.010$ | $\mathbf{0.597} \pm 0.058$ |
| $n_d = 5$ | $\mathbf{0.935} \pm 0.006$ | $\mathbf{0.600} \pm 0.035$ |
| | $\theta = 10^{-4}$ | $\theta = 10^{-4}$ |

similar to the case for $n_d = 3$. Redundant dampers provided the extra parameters, which sometimes made optimization easier, but were often ignored by learning identical properties to existing dampers, by adopting a zero damping coefficient, or by having zero coupling strength. Interestingly, this trend appeared in both the training and test subsets, which suggests that assessing performance on the training subset can help identify the correct number of dampers. In this way, PoDiNNs offer interpretable insights into the system's internal structure.

See Appendix G for additional visualizations and analyses.

## 5 CONCLUSION

In this study, we proposed *Poisson-Dirac Neural Networks* (PoDiNNs), which use a Dirac structure to unify the port-Hamiltonian and Poisson formulations. PoDiNNs offer a unified framework capable of handling various domains of dynamical systems, identifying internal coupling patterns, learning component-wise characteristics, and effectively modeling multiphysics systems. Our experiments with mechanical, rotational, electro-magnetic, and hydraulic systems validate these capabilities. Developing methods to address dynamical systems that are described by DAEs and partial differential equations remains for future research.

## ACKNOWLEDGMENTS

This study was partly supported by JST PRESTO (JPMJPR24TB, JPMJPR21C7), CREST (JP-MJCR1914, JPMJCR24Q5), ASPIRE (JPMJAP2329), Moonshot R&D (JPMJMS2033-14), and JSPS KAKENHI (24K15105), and partly achieved through the use of SQUID at D3 Center, Osaka University.

## ETHICS STATEMENT

This study is purely focused on dynamical systems and modeling, and it is not expected to have any direct negative impact on society or individuals.

## REPRODUCIBILITY STATEMENT

The environment, datasets, methods, evaluation metrics, and other experimental settings are given in Section 4.1 and Appendix F. For full reproducibility, it is recommended to run the source code attached as supplementary material.

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

# Appendix

## A    LESS RELATED WORK

Research on methods for learning dynamical systems has a long history (Nelles, 2001). Although the approaches discussed here are not directly related to the primary focus of this paper, we include them for completeness and to clarify key distinctions. Since linear systems are relatively straightforward to handle, our discussion is restricted to nonlinear systems.

A classical approach to learning nonlinear dynamics considers discrete-time systems of the form $x_{n+1} = f(x_n)$, where the mapping $f$ is approximated using a neural network (Nelles, 2001). Even when the underlying system is continuous in time, one can sample the trajectory at fixed intervals and treat it as a discrete-time dynamics. A variant of this approach models the dynamics as $x_{n+1} = x_n + \Phi(x_n)$, which closely resembles the forward Euler method. Building on this idea, Wang & Lin (1998) proposed a neural network architecture inspired by the fourth-order Runge-Kutta method. Neural ODEs can be regarded as an extension of this approach although the original work noted connections to ResNet (He et al., 2016).

If state $x$ is only partially observable, the dynamics is typically modeled as a combination of latent-state dynamics and an observation model $y = \phi(x)$, which is commonly referred to as a state-space model (SSM). Among neural architectures, long short-term memories (LSTMs) can be interpreted as discrete-time SSMs (Hochreiter & Schmidhuber, 1997). More sophisticated models combine neural networks with Bayesian inference to handle uncertainty in latent states (Krishnan et al., 2017; Becker et al., 2018). Recently, SSMs have been employed in sequence modeling tasks such as language processing, leading to architectures like S4 and Mamba (Gu et al., 2022; Gu & Dao, 2024). These models are primarily designed for modeling logical reasoning or expressing causal relationships rather than for modeling physical dynamical systems.

A different viewpoint on dynamical systems leverages the Koopman operator $\mathcal{K}$, which describes the time evolution of observations $y$ rather than the latent states $x$ themselves (Bevanda et al., 2021). Formally, the operator $\mathcal{K}$ is linear (albeit typically infinite-dimensional) and satisfies $y(t + s) = \mathcal{K}^s(y(t))$. The Koopman operator provides an alternative perspective for an analysis, rather than for modeling.

Another framework that is sometimes confused with our primary focus is that of Physics-Informed Neural Networks (PINNs) (Sirignano & Spiliopoulos, 2018; Raissi et al., 2019; Du & Zaki, 2021). PINNs are designed to *solve* known symbolic governing equations by employing neural networks as basis functions, analogous to classical numerical methods such as the finite element method. Although PINNs can be used to estimate certain parameters within symbolic equations from data, they are not intended for the discovery of unknown dynamical systems.

For directly learning symbolic governing equations from data, the Sparse Identification of Nonlinear Dynamics (SINDy) framework has been extensively investigated (Brunton et al., 2016). SINDy typically represents the dynamics as a combination of predefined terms in a dictionary, which introduces a trade-off between expressiveness and interpretability.

As summarized in Table 1, the research trend to which our work contributes is focused on data-driven modeling of dynamical systems. We assume that the governing symbolic equations are unknown and not the primary target for estimation, and that the full state of the system is observable. While combining the proposed method with the aforementioned approaches could be a promising direction for future research, it is not meaningful to directly compare their performance.

Moreover, our study specifically considers dynamical systems as coupled systems across multiple physical domains and focuses on learning and uncovering both the coupling patterns and the characteristics of individual components. This distinguishes our work from previous research. Although system identification for coupled linear systems has been extensively studied (Nelles, 2001), to the best of our knowledge, no prior work has tackled flexible identification using neural networks.

## B    EXAMPLES OF FORMULATIONS

Consider the time evolution of a point $\boldsymbol{u}$ on a manifold $\mathcal{M}$. If a local coordinate on $\mathcal{M}$ is denoted by $x_i$, the corresponding basis vector of the tangent space $T_{\boldsymbol{u}}\mathcal{M}$ is denoted by $\frac{\partial}{\partial x_i}$. A vector field $X$ on $\mathcal{M}$ is expressed as $X = \sum_i X_i \frac{\partial}{\partial x_i}$. If the curve $\boldsymbol{u}(t)$ satisfies $\dot{\boldsymbol{u}}(t) = X_{\boldsymbol{u}(t)}$, $X_i$ represents the local rate of change of the state $\boldsymbol{u}(t)$ in the $x_i$-direction, i.e., $\dot{u}_i = X_i$. For a function $F : \mathcal{M} \to \mathbb{R}$, its differential $\mathrm{d}F$ is given by $\mathrm{d}F = \sum_i \frac{\partial F}{\partial x_i} \mathrm{d}x_i$. The differential is a covector field, that is, a collection of points on cotangent spaces $T_{\boldsymbol{u}}^*\mathcal{M}$. The basis vector of the cotangent space $T_{\boldsymbol{u}}^*\mathcal{M}$ is $\mathrm{d}x_i$, and the following relation holds: $\mathrm{d}x_i \frac{\partial}{\partial x_j} = 1$ if $i = j$ and 0 otherwise.

When a function $F$ defines dynamics, a 2-tensor field links a vector field $X$ and the covector field $\mathrm{d}F$. Such a 2-tensor field can be a symplectic form, Poisson bivector, or Riemannian metric. For example, a link with the negative of a Riemannian metric defines a gradient flow. A symplectic form and Poisson bivector are defined using the wedge product $\wedge$, which satisfies the skew-symmetry $\mathrm{d}x_i \wedge \mathrm{d}x_j = -\mathrm{d}x_j \wedge \mathrm{d}x_i$, and the relation $(\mathrm{d}x_i \wedge \mathrm{d}x_j)(\frac{\partial}{\partial x_i}) = \mathrm{d}x_j$. Marsden & Ratiu (1999) and Hairer et al. (2006) have thoroughly discussed how to describe dynamical systems using these geometric objects. While theoretical details are left to these textbooks, this section introduces a mass-spring system as a concrete example.

### B.1    MASS-SPRING SYSTEM

Consider a mass-spring system with spring constant $k$ and mass $m$. We will write its dynamics by several formulations.

**Canonical Hamiltonian Systems**    In the Darboux coordinates (i.e., on the cotangent bundle $T^*\mathcal{Q}$), the generalized coordinate $q$ is the displacement of spring $k$, and the generalized momentum $p$ is obtained as $p = mv$ for the velocity $v$ of mass $m$. The manifold $\mathcal{M}$ is a 2-dimensional Euclidean space $\mathbb{R}^2$. The Hamiltonian $H$ is $H(q, p) = \frac{1}{2m}p^2 + \frac{1}{2}kq^2$, and the symplectic form $\Omega$ is standard, i.e., $\Omega = \mathrm{d}q \wedge \mathrm{d}p$.

Hamilton's equations state that $(\Omega^\flat)(X_H) = (\mathrm{d}q \wedge \mathrm{d}p)(\dot{q}\frac{\partial}{\partial q} + \dot{p}\frac{\partial}{\partial p}) = \dot{q}\mathrm{d}p - \dot{p}\mathrm{d}q$ equals $\mathrm{d}H = \frac{\partial H}{\partial q}\mathrm{d}q + \frac{\partial H}{\partial p}\mathrm{d}p$, leading to the equations of motion, $\dot{q} = \frac{\partial H}{\partial p} = p/m$, $\dot{p} = -\frac{\partial H}{\partial q} = -kq$.

**Non-Canonical Hamiltonian Systems**    On the tangent bundle $\mathcal{M} = T\mathcal{Q}$, the velocity $v$ is used in place of the momentum $p$, and the symplectic form $\Omega$ is the Lagrangian 2-form $\Omega = m(\mathrm{d}q \wedge \mathrm{d}v)$. The Hamiltonian $H$ is $H(q, v) = \frac{1}{2}mv^2 + \frac{1}{2}kq^2$.

Hamilton's equations state that $(\Omega^\flat)(X_H) = m(\mathrm{d}q \wedge \mathrm{d}v)(\dot{q}\frac{\partial}{\partial q} + \dot{v}\frac{\partial}{\partial v}) = m\dot{q}\mathrm{d}v - m\dot{v}\mathrm{d}q$ equals $\mathrm{d}H = \frac{\partial H}{\partial q}\mathrm{d}q + \frac{\partial H}{\partial v}\mathrm{d}v$, leading to the equations of motion, $\dot{q} = \frac{1}{m}\frac{\partial H}{\partial v} = v$, $\dot{v} = -\frac{1}{m}\frac{\partial H}{\partial q} = -kq/m$.

**Poisson Systems**    In the Darboux coordinates, the Poisson bivector $B$ is $B = \frac{\partial}{\partial p} \wedge \frac{\partial}{\partial q}$. Hamilton's equations state that $X_H = \dot{q}\frac{\partial}{\partial q} + \dot{p}\frac{\partial}{\partial p}$ equals $B^\sharp(\mathrm{d}H) = (\frac{\partial}{\partial p} \wedge \frac{\partial}{\partial q})(\frac{\partial H}{\partial q}\mathrm{d}q + \frac{\partial H}{\partial p}\mathrm{d}p) = \frac{\partial H}{\partial p}\frac{\partial}{\partial q} - \frac{\partial H}{\partial q}\frac{\partial}{\partial p}$. The equations of motion are $\dot{q} = \frac{\partial H}{\partial p} = p/m$, $\dot{p} = -\frac{\partial H}{\partial q} = -kq$.

On the tangent bundle $\mathcal{M} = T\mathcal{Q}$, the Poisson bivector $B$ is $B = \frac{1}{m}(\frac{\partial}{\partial v} \wedge \frac{\partial}{\partial q})$, and the Hamiltonian is $H(q, v) = \frac{1}{2}mv^2 + \frac{1}{2}kq^2$. Hamilton's equations state that $X_H = \dot{q}\frac{\partial}{\partial q} + \dot{v}\frac{\partial}{\partial v}$ equals $B^\sharp(\mathrm{d}H) = \frac{1}{m}(\frac{\partial}{\partial v} \wedge \frac{\partial}{\partial q})(\frac{\partial H}{\partial q}\mathrm{d}q + \frac{\partial H}{\partial v}\mathrm{d}v) = \frac{1}{m}\frac{\partial H}{\partial v}\frac{\partial}{\partial q} - \frac{1}{m}\frac{\partial H}{\partial q}\frac{\partial}{\partial v}$. The equations of motion are $\dot{q} = \frac{1}{m}\frac{\partial H}{\partial v} = v$, $\dot{v} = -\frac{1}{m}\frac{\partial H}{\partial q} = -kq/m$.

### B.2    CONSTRAINED MASS-SPRING SYSTEM

Consider a pair of mass-spring systems, indexed by $i \in \{1, 2\}$, and introduce a constraint such that the two masses are coupled and always have the same displacement and velocity; the dynamics is degenerate. This constraint is expressed as $q_1 = q_2$. The Hamiltonian is given simply by the sum of two systems, $H(\boldsymbol{q}, \boldsymbol{p}) = \frac{1}{2m_1}p_1^2 + \frac{1}{2m_2}p_2^2 + \frac{1}{2}k_1q_1^2 + \frac{1}{2}k_2q_2^2$. However, Hamilton's equations $\Omega^\flat(X_H) = \mathrm{d}H$ with the standard symplectic form $\Omega$ cannot describe the dynamics.

**Degenerate Systems with Coordinate Transformation**  Define new coordinates $q = q_1 = q_2$ and $p = \frac{m_1+m_2}{m_1}p_1 = \frac{m_1+m_2}{m_2}p_2$, and consider the submanifold $\tilde{\mathcal{M}}$ spanned by $(q,p)$. The Hamiltonian $H$ is unchanged but rewritten as $H(q,p) = \frac{1}{2(m_1+m_2)}p^2 + \frac{1}{2}(k_1+k_2)q^2$. With the standard symplectic form $\Omega = \mathrm{d}q \wedge \mathrm{d}p$ on $\tilde{\mathcal{M}}$, the equations of motion are written as $\dot{q} = \frac{p}{m_1+m_2}$, $\dot{p} = -(k_1+k_2)q$.

**Degenerate Systems with Constraint Force**  The constraint on the coordinates, $q_1 - q_2 = 0$, is naturally satisfied by the constraint on the velocities, $\dot{q}_1 - \dot{q}_2 = 0$, which is written as the constraint on momenta, $p_1/m_1 - p_2/m_2 = 0$. The constants of motion are $q_1 - q_2$ and $p_1/m_1 - p_2/m_2$. A constraint force can be introduced to satisfy these constraints, yielding the same equation as above. Finzi et al. (2020) proposed CHNNs by combining this approach with neural networks. However, the automatic derivation of the constraint on momenta from that on the coordinates is non-trivial, and it remains unclear how to learn constraints from data.

**Degenerate Systems as Poisson Systems**  The equations of motion on the submanifold $\tilde{\mathcal{M}}$, $\dot{q} = \frac{p}{m_1+m_2}$, $\dot{p} = -(k_1+k_2)q$, can be rewritten in the original coordinate system as $\dot{q}_1 = \frac{p_1+p_2}{m_1+m_2}$, $\dot{q}_2 = \frac{p_1+p_2}{m_1+m_2}$, $\dot{p}_1 = -\frac{m_1}{m_1+m_2}(k_1q_1+k_2q_2)$, $\dot{p}_2 = -\frac{m_2}{m_1+m_2}(k_1q_1+k_2q_2)$. Even on the original manifold $\mathcal{M}$, these equations are obtained from Hamilton's equations $X = B^\sharp(\mathrm{d}H)$ with the Poisson bivector $B = \frac{1}{m_1+m_2}(m_1\frac{\partial}{\partial p_1} + m_2\frac{\partial}{\partial p_2}) \wedge (\frac{\partial}{\partial q_1} + \frac{\partial}{\partial q_2})$ on $\mathcal{M}$, while its bundle map $B^\sharp$ is degenerate. This fact implies that, by adjusting the Poisson bivector from data, we can learn the Hamiltonian systems with constraints and identify how the dynamics is degenerate.

On the tangent bundle $\mathcal{M} = T\mathcal{Q}$, the Hamiltonian is $H(q_1,q_2,v_1,v_2) = \frac{1}{2}m_1v_1^2 + \frac{1}{2}m_2v_2^2 + \frac{1}{2}k_1q_1^2 + \frac{1}{2}k_2q_2^2$, and the Poisson bivector $B$ is $B = \frac{1}{m_1+m_2}(\frac{\partial}{\partial v_1} + \frac{\partial}{\partial v_2}) \wedge (\frac{\partial}{\partial q_1} + \frac{\partial}{\partial q_2})$. Then, the equations of motion are $\dot{q}_1 = \frac{m_1v_1+m_2v_2}{m_1+m_2}$, $\dot{q}_2 = \frac{m_1v_1+m_2v_2}{m_1+m_2}$, $\dot{v}_1 = -\frac{k_1q_1+k_2q_2}{m_1+m_2}$, $\dot{v}_2 = -\frac{k_1q_1+k_2q_2}{m_1+m_2}$.

## C  POISSON SYSTEMS

### C.1  MECHANICAL SYSTEMS

Using the mass-spring systems described above, we provide concrete examples of how the bivector represents coupling and constraints in mechanical systems.

**Coupled Systems as Poisson Systems**  Consider two masses and two springs, indexed by $i \in \{1,2\}$, coupled in sequence from a fixed wall. Let $q_i$ denote the displacement of $i$-th spring. The equations of motion are given by $\dot{q}_1 = p_1/m_1$, $\dot{q}_2 = p_2/m_2 - p_1/m_1$, $\dot{p}_1 = -k_1q_1 + k_2q_2$, and $\dot{p}_2 = -k_2q_2$. The bivector $B$ leading to these equations is $B = \frac{\partial}{\partial p_1} \wedge \frac{\partial}{\partial q_1} - \frac{\partial}{\partial p_1} \wedge \frac{\partial}{\partial q_2} + \frac{\partial}{\partial p_2} \wedge \frac{\partial}{\partial q_2}$ for the Hamiltonian $H(\boldsymbol{q},\boldsymbol{p}) = \frac{1}{2m_1}p_1^2 + \frac{1}{2m_2}p_2^2 + \frac{1}{2}k_1q_1^2 + \frac{1}{2}k_2q_2^2$. These terms indicate the couplings between $p_1$ and $q_1$, $p_1$ and $q_2$, and $p_2$ and $q_2$. The negative coefficient indicates that the coupling between $p_1$ and $q_2$ is in the opposite direction.

**Degenerate Systems as Poisson Systems**  As shown in Appendix B.2, degenerate systems can be expressed as Poisson systems. Recall the case of a pair of mass-spring systems, indexed by $i \in \{1,2\}$, constrained so that the two masses always have the same displacement and velocity. This system is expressed with the Poisson bivector $B = \frac{1}{m_1+m_2}(m_1\frac{\partial}{\partial p_1} + m_2\frac{\partial}{\partial p_2}) \wedge (\frac{\partial}{\partial q_1} + \frac{\partial}{\partial q_2})$, which is degenerate in the sense that its bundle map $B^\sharp$ is degenerate. This indicates the absence of a corresponding symplectic form. Conversely, by learning the bivector $B$ and examining how it degenerates, one can identify the constraints imposed on the system.

The coefficient $\frac{m_1}{m_1+m_2}$ for the term $\frac{\partial}{\partial p_1} \wedge \frac{\partial}{\partial q_1}$ indicates the coupling strength between mass $p_1$ and spring $q_1$. The effort $e_{q_1}^S = \frac{\partial H}{\partial q_1}$ from spring $q_1$ is distributed to the masses such that $\frac{m_1}{m_1+m_2}e_{q_1}$ goes to mass $m_1$ and $\frac{m_2}{m_1+m_2}e_{q_1}$ goes to mass $m_2$.

**Coordinate Transformation by Bivector**  As shown in Appendix B.2, the elements of the Poisson bivector $B$ in a Poisson system depend on whether the dynamics are defined on the tangent bundle $T\mathcal{Q}$ (using generalized velocities as part of the state) or the cotangent bundle $T^*\mathcal{Q}$ (using generalized momenta as part of the state). Despite this difference, both coordinate systems are represented by Poisson bivectors. Thus, by learning the bivector $B$ to approximate the dynamics of a given system, one can implicitly learn the coordinate system employed by that system.

## C.2 ELECTRIC CIRCUITS

Our formulation is applicable to electric circuits, as summarized in Table 2. Let $I_X$ and $V_X$ denote the current through and voltage across a circuit element $X$, respectively.

A capacitor with capacitance $C$ has the electric charge $Q$ as its state, and stores the energy $H_C = \frac{1}{2C}Q^2$. Its flow is the current $I_C$, which leads to the change in the electric charge $Q$ as $\dot{Q} = I_C$. Its effort is the voltage $V_C$ because the stored electric charge $Q$ generates the voltage $V_C$ as $V_C = \frac{Q}{C} = \frac{\partial H_C}{\partial Q}$.

An inductor with inductance $L$ has the magnetic flux $\varphi$ as its state, and stores the energy $H_L = \frac{1}{2L}\varphi^2$. Its flow is the voltage $V_L$, which leads to the change in the magnetic flux $\varphi$ as $\dot{\varphi} = V_L$. Its effort is the current $I_L$ because the magnetic flux $\varphi$ generates the current $I_L$ as $I_L = \frac{\varphi}{L} = \frac{\partial H_L}{\partial \varphi}$.

Consider a system composed of an inductor $L$ and a capacitor $C$ coupled in series. The state space $\mathcal{M}$ is the space of the magnetic flux $\varphi$ and electric charge $Q$. The total energy is $H = H_L + H_C$, and its differential is $\mathrm{d}H = \frac{\varphi}{L}\mathrm{d}\varphi + \frac{Q}{C}\mathrm{d}Q$. Define a bivector $B = \frac{\partial}{\partial \varphi} \wedge \frac{\partial}{\partial Q}$, which leads to

$$B^\sharp(\mathrm{d}H) = \frac{\varphi}{L}\frac{\partial}{\partial Q} - \frac{Q}{C}\frac{\partial}{\partial \varphi}.$$

The vector field on $\mathcal{M}$ is $X = \dot{\varphi}\frac{\partial}{\partial \varphi} + \dot{Q}\frac{\partial}{\partial Q}$. Hamilton's equations $X = B^\sharp(\mathrm{d}H)$ lead to the equations of motion:

$$V_L = \dot{\varphi} = -\frac{Q}{C} = -V_C \text{ and } I_C = \dot{Q} = \frac{\varphi}{L} = I_L.$$

However, electrical circuits that can be described as Poisson systems are limited to energy-conservative LC circuits. Our formulation extends this to include resistors, diodes, voltage sources, and current sources.

# D DIRAC STRUCTURE

## D.1 DIRAC STRUCTURE ON A VECTOR SPACE

*Proof of Theorem 1(Yoshimura & Marsden, 2006a).* By Definition 1,

$$D_V^\perp = \{(\boldsymbol{w}, \boldsymbol{\beta}) \in V \times V^* \mid \langle \boldsymbol{\alpha}, \boldsymbol{w} \rangle + \langle \boldsymbol{\beta}, \boldsymbol{v} \rangle = 0 \text{ for all } \boldsymbol{v} \in \Delta \text{ and } \boldsymbol{\alpha} \in \Delta^\circ\}.$$

Let $(\bar{\boldsymbol{v}}, \bar{\boldsymbol{\alpha}}) \in D_V$. Then, $\bar{\boldsymbol{v}} \in \Delta$ and $\bar{\boldsymbol{\alpha}} \in \Delta^\circ$, so $\langle \boldsymbol{\alpha}, \bar{\boldsymbol{v}} \rangle + \langle \bar{\boldsymbol{\alpha}}, \boldsymbol{v} \rangle = 0$ for all $(\boldsymbol{v}, \boldsymbol{\alpha}) \in D_V$. This implies $(\bar{\boldsymbol{v}}, \bar{\boldsymbol{\alpha}}) \in D_V^\perp$ Thus, $D_V \subset D_V^\perp$.

Let $(\boldsymbol{w}, \boldsymbol{\beta}) \in D_V^\perp$. In the above definition of $D_V^\perp$, setting $\boldsymbol{\alpha} = 0$ gives $\langle \boldsymbol{\beta}, \boldsymbol{v} \rangle = 0$ for all $\boldsymbol{v} \in \Delta$. Hence, $\boldsymbol{\beta} \in \Delta^\circ$. Similarly, setting $\boldsymbol{v} = \boldsymbol{0}$ gives $\langle \boldsymbol{\alpha}, \boldsymbol{w} \rangle = 0$ for all $\boldsymbol{\alpha} \in \Delta^\circ$, which implies $\boldsymbol{w} \in \Delta$. Hence, $(\boldsymbol{w}, \boldsymbol{\beta}) \in D_V$. Thus, $D_V^\perp \subset D_V$.

Therefore, $D_V = D_V^\perp$. $\qquad\square$

## D.2 DIRAC STRUCTURE ON A MANIFOLD

**Definition 4** (Courant (1990); Yoshimura & Marsden (2006a)). *A distribution $\Delta$ on a manifold $\mathcal{M}$ is a collection of vector subspaces $\Delta_{\boldsymbol{u}}$ of tangent spaces $T_{\boldsymbol{u}}\mathcal{M}$, each assigned smoothly to $\mathcal{M}$ at point $\boldsymbol{u}$, forming a vector subbundle of $T\mathcal{M}$. The annihilator $\Delta^\circ$ of $\Delta$ is a collection of the annihilator $\Delta_{\boldsymbol{u}}^\circ$ of $\Delta_{\boldsymbol{u}}$, also forming a subbundle of $T^*\mathcal{M}$. Then, a Dirac structure $D$ is constructed as $D = \Delta \oplus \Delta^\circ$, which is a subbundle of the Pontryagin bundle $T\mathcal{M} \oplus T^*\mathcal{M}$.*

A type of Dirac structure on a manifold $\mathcal{M}$ can be defined using a symplectic form $\Omega$ on $\mathcal{M}$.

**Theorem 3** (Yoshimura & Marsden (2006a)). *Let $\Delta_{\mathcal{M}}$ be a distribution on a manifold $\mathcal{M}$. Let $\Omega$ be a symplectic form on a manifold $\mathcal{M}$. $\Omega$ is restricted to $\Delta_{\mathcal{M}}$ and denoted by $\Omega_{\Delta_{\mathcal{M}}}$. Then, the collection of*

$$(D_{\Delta_{\mathcal{M}}})_{\boldsymbol{u}} = \{(\boldsymbol{v}, \boldsymbol{\alpha}) \in T_{\boldsymbol{u}}\mathcal{M} \times T_{\boldsymbol{u}}^*\mathcal{M} \mid \boldsymbol{v} \in (\Delta_{\mathcal{M}})_{\boldsymbol{u}},$$
$$\text{and } \langle \boldsymbol{\alpha}, \boldsymbol{w} \rangle = (\Omega_{\Delta_{\mathcal{M}}})_{\boldsymbol{u}}(\boldsymbol{v}, \boldsymbol{w}) \text{ for all } \boldsymbol{w} \in (\Delta_{\mathcal{M}})_{\boldsymbol{u}}\}$$

*is a Dirac structure $D_{\Delta_\mathcal{M}} \subset T\mathcal{M} \oplus T^*\mathcal{M}$ on $\mathcal{M}$, which is said to be induced by $\Omega$.*

*Proof of Theorem 3(Yoshimura & Marsden, 2006a).* By definition, the orthogonal of $D_{\Delta_\mathcal{M}}$ at $\boldsymbol{u} \in \mathcal{M}$ is given by

$$(D_{\Delta_\mathcal{M}}^\perp)_{\boldsymbol{u}} = \{(\boldsymbol{w}, \boldsymbol{\beta}) \in T_{\boldsymbol{u}}\mathcal{M} \times T_{\boldsymbol{u}}^*\mathcal{M} \mid \langle \boldsymbol{\alpha}, \boldsymbol{w} \rangle + \langle \boldsymbol{\beta}, \boldsymbol{v} \rangle = 0 \text{ for all } \boldsymbol{v} \in (\Delta_\mathcal{M})_{\boldsymbol{u}},$$
$$\text{and } \langle \boldsymbol{\alpha}, \boldsymbol{w} \rangle = (\Omega_{\Delta_\mathcal{M}})_{\boldsymbol{u}}(\boldsymbol{v}, \boldsymbol{w}) \text{ for all } \boldsymbol{w} \in (\Delta_\mathcal{M})_{\boldsymbol{u}}\}.$$

Let $(\boldsymbol{v}, \boldsymbol{\alpha}), (\bar{\boldsymbol{v}}, \bar{\boldsymbol{\alpha}}) \in (D_{\Delta_\mathcal{M}})_{\boldsymbol{u}}, \langle \boldsymbol{\alpha}, \bar{\boldsymbol{v}} \rangle + \langle \bar{\boldsymbol{\alpha}}, \boldsymbol{v} \rangle = \Omega_{\Delta_\mathcal{M}}(\boldsymbol{v}, \bar{\boldsymbol{v}}) + \Omega_{\Delta_\mathcal{M}}(\bar{\boldsymbol{v}}, \boldsymbol{v}) = 0$. The latter equality holds because of the skew-symmetry of $\Omega$. This implies $(\boldsymbol{v}, \boldsymbol{\alpha}) \in (D_{\Delta_\mathcal{M}}^\perp)_{\boldsymbol{u}}$. Thus, $(D_{\Delta_\mathcal{M}})_{\boldsymbol{u}} \subset (D_{\Delta_\mathcal{M}}^\perp)_{\boldsymbol{u}}$.

Let $(\boldsymbol{w}, \boldsymbol{\beta}) \in (D_{\Delta_\mathcal{M}}^\perp)_{\boldsymbol{u}}$. Then, $\langle \boldsymbol{\alpha}, \boldsymbol{w} \rangle + \langle \boldsymbol{\beta}, \boldsymbol{v} \rangle = 0$ for all $(\boldsymbol{v}, \boldsymbol{\alpha}) \in (D_{\Delta_\mathcal{M}})_{\boldsymbol{u}}$. First, setting $\boldsymbol{v} = \boldsymbol{0}$ gives $\langle \boldsymbol{\alpha}, \boldsymbol{w} \rangle = (\Omega_{\Delta_\mathcal{M}})_{\boldsymbol{u}}(\boldsymbol{0}, \boldsymbol{w}) = 0$ for any $\boldsymbol{\alpha} \in (\Delta_\mathcal{M}^\circ)_{\boldsymbol{u}}$. Thus, $\boldsymbol{w} \in (\Delta_\mathcal{M}^\circ)_{\boldsymbol{u}}^\circ = (\Delta_\mathcal{M})_{\boldsymbol{u}}$. Second, if $\boldsymbol{\alpha}$ satisfies $\langle \boldsymbol{\alpha}, \boldsymbol{w} \rangle = (\Omega_{\Delta_\mathcal{M}})_{\boldsymbol{u}}(\boldsymbol{v}, \boldsymbol{w})$ for any $\boldsymbol{v} \in (\Delta_\mathcal{M})_{\boldsymbol{u}}$, then $\langle \boldsymbol{\alpha}, \boldsymbol{w} \rangle + \langle \boldsymbol{\beta}, \boldsymbol{v} \rangle = (\Omega_{\Delta_\mathcal{M}})_{\boldsymbol{u}}(\boldsymbol{v}, \boldsymbol{w}) + \langle \boldsymbol{\beta}, \boldsymbol{v} \rangle = 0$ for any $\boldsymbol{v} \in (\Delta_\mathcal{M})_{\boldsymbol{u}}$. This implies $\langle \boldsymbol{\beta}, \boldsymbol{v} \rangle = (\Omega_{\Delta_\mathcal{M}})_{\boldsymbol{u}}(\boldsymbol{w}, \boldsymbol{v})$ for any $\boldsymbol{v} \in (\Delta_\mathcal{M})_{\boldsymbol{u}}$. Because it has been proved that $\boldsymbol{w} \in (\Delta_\mathcal{M})_{\boldsymbol{u}}, (\boldsymbol{w}, \boldsymbol{\beta}) \in (D_{\Delta_\mathcal{M}})_{\boldsymbol{u}}$. Thus, $(D_{\Delta_\mathcal{M}}^\perp)_{\boldsymbol{u}} \subset (D_{\Delta_\mathcal{M}})_{\boldsymbol{u}}$.

Therefore, $(D_{\Delta_\mathcal{M}})_{\boldsymbol{u}} = (D_{\Delta_\mathcal{M}}^\perp)_{\boldsymbol{u}}$. $\qquad\square$

**Definition 5** (Hamilton-Dirac System (Yoshimura & Marsden, 2006b)). *Let $\mathcal{M}$ be a manifold, $H : \mathcal{M} \to \mathbb{R}$ be an energy function, and $D$ be a Dirac structure on $\mathcal{M}$. Given a vector field $X_H$ on $\mathcal{M}$, if it holds for each $\boldsymbol{u}(t) \in \mathcal{M}$ and $t \in [a, b]$ that*

$$((X_H)_{\boldsymbol{u}(t)}, \mathrm{d}H_{\boldsymbol{u}(t)}) \in (D_{\Delta_\mathcal{M}})_{\boldsymbol{u}(t)},$$

*the tuple $(H, D_{\Delta_\mathcal{M}}, X_H)$ is called a Hamilton-Dirac system (or implicit Hamiltonian system).*

Note that the symplectic form restricted to a distribution $\Delta_\mathcal{M}$, denoted by $\Omega_{\Delta_\mathcal{M}}$, satisfies $(\Omega_{\Delta_\mathcal{M}})_{\boldsymbol{u}}(\boldsymbol{v}, \boldsymbol{w}) = \Omega_{\boldsymbol{u}}(\boldsymbol{v}, \boldsymbol{w})$ for any $\boldsymbol{v}, \boldsymbol{w} \in (\Delta_\mathcal{M})_{\boldsymbol{u}}$. The symplectic form $\Omega$ is degenerate in the sense that its bundle map $\Omega^\flat$ on the tangent space $T_{\boldsymbol{u}}\mathcal{M}$ is degenerate, but $\Omega_{\Delta_\mathcal{M}}$ is non-degenerate on the distribution $(\Delta_\mathcal{M})_{\boldsymbol{u}}$. If there is no constraint, $\Delta_\mathcal{M} = T\mathcal{M}$ and $\Omega_{\Delta_\mathcal{M}} = \Omega$. When the Dirac structure $D$ is given as in Theorem 3 with no constraint, the Hamilton-Dirac system is identical to the Hamiltonian system $\Omega^\flat(X_H) = \mathrm{d}H$ in Eq. (2). The distribution $\Delta_\mathcal{M}$ describes constraints on the velocities, and hence constrained Hamiltonian systems can be rewritten as Hamilton-Dirac systems. Therefore, Hamiltonian-Dirac systems are generalizations of Hamiltonian and constrained Hamiltonian systems. For example, the system in Appendix B.2 can be written as a Hamilton-Dirac system as follows:

**Degenerate Systems as Hamilton-Dirac Systems** The distribution $\Delta_\mathcal{M}$ is given by $(\Delta_\mathcal{M})_{\boldsymbol{u}} = \{(\dot{q}_1, \dot{q}_2, \dot{p}_1, \dot{p}_2) \in T_{\boldsymbol{u}}\mathcal{M} \mid \dot{q}_1 - \dot{q}_2 = 0, \ \dot{p}_1/m_1 - \dot{p}_2/m_2 = 0\} = \mathrm{span}\{\frac{\partial}{\partial q_1} + \frac{\partial}{\partial q_2}, \frac{1}{m_1}\frac{\partial}{\partial p_1} + \frac{1}{m_2}\frac{\partial}{\partial p_2}\}$. Define the new coordinates $q = q_1 = q_2$ and $p = \frac{m_1+m_2}{m_1}p_1 = \frac{m_1+m_2}{m_2}p_2$, with $\Delta_\mathcal{M} = \mathrm{span}\{\frac{\partial}{\partial q}, \frac{\partial}{\partial p}\}$. The symplectic form $\Omega_{\Delta_\mathcal{M}}$ restricted to the distribution $\Delta_\mathcal{M}$ should satisfy $(\Omega_{\Delta_\mathcal{M}})_{\boldsymbol{u}}(\boldsymbol{v}, \boldsymbol{w}) = \Omega_{\boldsymbol{u}}(\boldsymbol{v}, \boldsymbol{w})$ for any $\boldsymbol{v}, \boldsymbol{w} \in (\Delta_\mathcal{M})_{\boldsymbol{u}}$ at $\boldsymbol{u}$. Such form is given as $\Omega_{\Delta_\mathcal{M}} = \mathrm{d}q \wedge \mathrm{d}p$. Then, the equations of motion are written as $\dot{q} = \frac{p}{m_1+m_2}, \ \dot{p} = -(k_1 + k_2)q$.

## D.3 PORT-BASED SYSTEMS

*Proof of Theorem 2.* By definition, the orthogonal of $D$ at $\boldsymbol{u} \in \mathcal{M}$ is given by

$$D_{\boldsymbol{u}}^\perp = \{(\bar{\boldsymbol{f}}, \bar{\boldsymbol{e}}) \in \mathcal{F}_{\boldsymbol{u}} \times \mathcal{E}_{\boldsymbol{u}} \mid \langle \boldsymbol{e}, \bar{\boldsymbol{f}} \rangle + \langle \bar{\boldsymbol{e}}, \boldsymbol{f} \rangle = 0 \text{ for all } \boldsymbol{e} \in \mathcal{E}_{\boldsymbol{u}} \text{ and } \boldsymbol{f} = B_{\boldsymbol{u}}^\sharp(\boldsymbol{e})\}.$$

Let $(\boldsymbol{f}, \boldsymbol{e}), (\bar{\boldsymbol{f}}, \bar{\boldsymbol{e}}) \in D_{\boldsymbol{u}}$. Then, $\langle \boldsymbol{e}, \bar{\boldsymbol{f}} \rangle + \langle \bar{\boldsymbol{e}}, \boldsymbol{f} \rangle = \langle \boldsymbol{e}, B_{\boldsymbol{u}}^\sharp(\bar{\boldsymbol{e}}) \rangle + \langle \bar{\boldsymbol{e}}, B_{\boldsymbol{u}}^\sharp(\boldsymbol{e}) \rangle = B_{\boldsymbol{u}}(\boldsymbol{e}, \bar{\boldsymbol{e}}) + B_{\boldsymbol{u}}(\bar{\boldsymbol{e}}, \boldsymbol{e}) = 0$ due to the skew-symmetry of $B$. This implies $(\boldsymbol{f}, \boldsymbol{e}) \in D_{\boldsymbol{u}}^\perp$. Thus, $D_{\boldsymbol{u}} \subset D_{\boldsymbol{u}}^\perp$.

Let $(\bar{\boldsymbol{f}}, \bar{\boldsymbol{e}}) \in D_{\boldsymbol{u}}^\perp$. Then, for any $(\boldsymbol{f}, \boldsymbol{e}) \in D_{\boldsymbol{u}}$ (i.e., for any $\boldsymbol{e} \in \mathcal{E}_{\boldsymbol{u}}$ and $\boldsymbol{f} = B_{\boldsymbol{u}}^\sharp(\boldsymbol{e})$), $0 = \langle \boldsymbol{e}, \bar{\boldsymbol{f}} \rangle + \langle \bar{\boldsymbol{e}}, \boldsymbol{f} \rangle = \langle \boldsymbol{e}, \bar{\boldsymbol{f}} \rangle + \langle \bar{\boldsymbol{e}}, B_{\boldsymbol{u}}^\sharp(\boldsymbol{e}) \rangle = \langle \boldsymbol{e}, \bar{\boldsymbol{f}} \rangle + B_{\boldsymbol{u}}(\bar{\boldsymbol{e}}, \boldsymbol{e}) = \langle \boldsymbol{e}, \bar{\boldsymbol{f}} \rangle - B_{\boldsymbol{u}}(\boldsymbol{e}, \bar{\boldsymbol{e}}) = \langle \boldsymbol{e}, \bar{\boldsymbol{f}} \rangle - \langle \boldsymbol{e}, B_{\boldsymbol{u}}^\sharp(\bar{\boldsymbol{e}}) \rangle$. $\langle \boldsymbol{e}, \bar{\boldsymbol{f}} \rangle = B_{\boldsymbol{u}}(\boldsymbol{e}, \bar{\boldsymbol{e}})$ for all $\boldsymbol{e} \in \mathcal{E}_{\boldsymbol{u}}$ implies that $\bar{\boldsymbol{f}} = B_{\boldsymbol{u}}^\sharp(\bar{\boldsymbol{e}})$. Thus, $D_{\boldsymbol{u}}^\perp \subset D_{\boldsymbol{u}}$.

Therefore, $D_{\boldsymbol{u}} = D_{\boldsymbol{u}}^\perp$. $\qquad\square$

By extending the above Hamilton-Dirac system from the Pontryagin bundle $T\mathcal{M} \oplus T^*\mathcal{M}$ to the vector bundle $\mathcal{F} \oplus \mathcal{E}$, we obtain the so-called port-Hamiltonian systems (Courant, 1990; van der Schaft & Jeltsema, 2014). However, previous neural network-based methods employed the canonical form in Eq. (4) and have not attempted to identify coupling patterns or component-wise characteristics (Zhong et al., 2020; Eidnes et al., 2023; Neary & Topcu, 2023; Di Persio et al., 2024).

In a bond graph representation of a dynamical system, a bond represents a component, its ports (flow and effort) represent points of interaction with other bonds, and the arrows connected to the ports represent the interactions between them. Both the Port-Hamiltonian and our formulations, as well as their terminology, are based on this bond graph structure (Duindam et al., 2009).

Instead of the symplectic form $\Omega$, our formulation employs (Poisson) bivector $B$ to define the Dirac structure $D$ on the tangent and cotangent bundles ($T_{\boldsymbol{u}}\mathcal{M} = \mathcal{F}^S$ and $T^*_{\boldsymbol{u}}\mathcal{M} = \mathcal{E}^S$) plus the vector bundles for port variables ($\mathcal{F}^R$, $\mathcal{F}^I$, $\mathcal{E}^R$, and $\mathcal{E}^I$). As is the case with the symplectic form $\Omega$, we can constrain a Poisson bivector $B$ on a codistribution $\Delta^*_{\mathcal{M}}$, which is a subbundle of the cotangent bundle $T^*\mathcal{M}$, and then define the Dirac structure. However, because the bivector inherently handles constraints, we did not adopt this formulation. Thus, our formulation is a special case of the Poisson-Dirac formulation with ports, as summarized in Table 1. Note that, for a bivector to be called a Poisson bivector, it must satisfy certain conditions, such as the Jacobi identity (Courant, 1990). However, the learned bivector in our formulation does not necessarily satisfy these conditions, nor are these conditions required for the proofs presented earlier. Thus, throughout this paper, we refer to $B$ simply as a bivector, without restricting it to being a Poisson bivector.

## E  IMPLEMENTATION

### E.1  COORDINATE-BASED FORM AND REDUCTION TO EXISTING MODELS

In the main body, we have introduced PoDiNNs in a coordinate-free form for a general and theoretical derivation. However, by introducing a specific coordinate system, we can rewrite PoDiNNs in a more intuitive form at the cost of generality. In this section, we will clarify the relationship with previous work through such rewrite.

As stated in Theorem 2, PoDiNNs use a bivector $B$ to define the Dirac structure $D$ as a subbundle of $\mathcal{F} \oplus \mathcal{E}$. When local coordinates $x_i$ are introduced on $\mathcal{M}$, the corresponding basis vectors for the tangent space $T_{\boldsymbol{u}}\mathcal{M}$ (that is, $\mathcal{F}^S_{\boldsymbol{u}}$) are $\frac{\partial}{\partial x_i}$. Additionally, the basis vectors for $\mathcal{F}^R_{\boldsymbol{u}}$ and $\mathcal{F}^I_{\boldsymbol{u}}$ are specified as $\xi^R_i$ and $\xi^I_i$, respectively. The basis vectors for $\mathcal{E}_{\boldsymbol{u}}$ are then defined as their duals. The bivector $B$ at $\boldsymbol{u}$ can be expressed as a linear combination of wedge products of pairs of basis vectors for $\mathcal{F}_{\boldsymbol{u}} = \mathcal{F}^S_{\boldsymbol{u}} \oplus \mathcal{F}^R_{\boldsymbol{u}} \oplus \mathcal{F}^I_{\boldsymbol{u}}$.

We assume that the state space $\mathcal{M}$ is a Euclidean space. Then, at a specific point $\boldsymbol{u}$, flows $\boldsymbol{f}$ and efforts $\boldsymbol{e}$ can be represented as vectors, and the linear bundle map $B^\sharp_{\boldsymbol{u}}$ is represented as a skew-symmetric matrix $B^\sharp_{\boldsymbol{u}} \in \mathbb{R}^{m \times m}$, where $m$ denotes the total number of dimensions of $\mathcal{F}_{\boldsymbol{u}}$ (Marsden & Ratiu, 1999). A skew-symmetric matrix $M$ satisfies the property $M = -M^\top$, which implies that the maximum number of independent degrees of freedom is $\frac{1}{2}m(m-1)$. This characteristic arises from the skew-symmetry of the wedge product, particularly the relation $\frac{\partial}{\partial x_j} \wedge \frac{\partial}{\partial x_i} = -\frac{\partial}{\partial x_i} \wedge \frac{\partial}{\partial x_j}$. Each bivector element corresponds to a specific pair of elements within this matrix.

In this context, the equation $\boldsymbol{f} = B^\sharp_{\boldsymbol{u}}(\boldsymbol{e})$ in Theorem 2 can be expressed with a matrix-vector product:

$$\begin{bmatrix} \boldsymbol{f}^S \\ \boldsymbol{f}^R \\ \boldsymbol{f}^I \end{bmatrix} = B^\sharp_{\boldsymbol{u}} \begin{bmatrix} \boldsymbol{e}^S \\ \boldsymbol{e}^R \\ \boldsymbol{e}^I \end{bmatrix}.$$

By substituting the variables as in Definition 3, we obtain:

$$\begin{bmatrix} \dot{\boldsymbol{u}} \\ \boldsymbol{f}^R \\ \boldsymbol{f}^I \end{bmatrix} = B^\sharp_{\boldsymbol{u}} \begin{bmatrix} \nabla_{\boldsymbol{u}} H(\boldsymbol{u}) \\ R_{\boldsymbol{u}}(\boldsymbol{f}^R) \\ \boldsymbol{e}^I(t) \end{bmatrix},$$

where the differential $\mathrm{d}H$ reduces to the gradient $\nabla_{\boldsymbol{u}} H$ in a Euclidean space.

In the absence of dissipation ($R$) and external inputs ($I$), the equation reduces to

$$\dot{\boldsymbol{u}} = B_{\boldsymbol{u}}^{\sharp} \nabla_{\boldsymbol{u}} H(\boldsymbol{u}).$$

This form aligns with those derived in NSFs and CHNNs, up to certain constraints imposed on the matrix (Chen et al., 2021; Finzi et al., 2020). In this sense, PoDiNNs are a generalization of these previous models. Since NSFs are a generalization of LNNs, PoDiNNs are also a generalization of LNNs.

Furthermore, if the Darboux coordinates are introduced on $\mathcal{M}$, the state $\boldsymbol{u}$ can be decomposed into the position $\boldsymbol{q}$ and momentum $\boldsymbol{p}$, and the bivector $B$ takes the standard form $\sum_i \frac{\partial}{\partial p_i} \wedge \frac{\partial}{\partial q_i}$. The equation reduces to:

$$\begin{bmatrix} \dot{\boldsymbol{q}} \\ \dot{\boldsymbol{p}} \end{bmatrix} = \begin{bmatrix} O & I \\ -I & O \end{bmatrix} \begin{bmatrix} \nabla_{\boldsymbol{q}} H(\boldsymbol{q}, \boldsymbol{p}) \\ \nabla_{\boldsymbol{p}} H(\boldsymbol{q}, \boldsymbol{p}) \end{bmatrix},$$

where $\boldsymbol{u} = \begin{bmatrix} \boldsymbol{q} \\ \boldsymbol{p} \end{bmatrix}$. This is identical to Hamilton's equations in Eq. (1), which is the form HNNs employed. Therefore, PoDiNNs are a generalization of HNNs. If the target system is known to be learnable by HNNs, it is also learnable by PoDiNNs of the form identical to HNNs. Consequently, a performance comparison between PoDiNNs and HNNs is not meaningful.

Let us consider a specific example from Appendix C.2: a system where a capacitor $C$ and an inductor $L$ are coupled in series. The state of the capacitor $C$ is its electric charge $Q$, while the state of the inductor $L$ is its magnetic flux $\varphi$. Their coupling is represented by the bivector $B = \frac{\partial}{\partial \varphi} \wedge \frac{\partial}{\partial Q}$. Using local coordinates $(Q, \varphi)$, we can represent $B$ as the matrix $B = \begin{bmatrix} 0 & 1 \\ -1 & 0 \end{bmatrix}$. The Poisson-Dirac formulation states that applying this matrix to the effort $(\nabla_Q H_C, \nabla_\varphi H_L)$ yields the flows $(\dot{Q}, \dot{\varphi})$, that is,

$$\begin{bmatrix} I_C \\ V_L \end{bmatrix} = \begin{bmatrix} 0 & 1 \\ -1 & 0 \end{bmatrix} \begin{bmatrix} \nabla_Q H_C \\ \nabla_\varphi H_L \end{bmatrix} = \begin{bmatrix} I_L \\ -V_C \end{bmatrix}.$$

This corresponds to the standard form of Hamilton's equations. We can see that the first line represents Kirchhoff's current law, $I_C = I_L$, and the second line represents Kirchhoff's voltage law, $V_L = -V_C$. The sign inversion follows from the skew-symmetric property of the bivector $B$. As the number of components increases, the dimensionality of the flows, efforts, and matrix also grows, and the bivector may not retain a standard form, as observed in system (e). Nonetheless, the bivector consistently represents Kirchhoff's laws, independent of the component-wise characteristics, and the Poisson-Dirac formulation ensures the satisfaction of these laws. Similarly, in mechanical systems, bivectors represent the law of action and reaction.

In general, the coordinate systems are not limited to the Darboux coordinates, components can exhibit a variety of coupling patterns, and the matrix $B_{\boldsymbol{u}}^{\sharp}$ can contain elements beyond the restricted values of -1, 0, or 1. The state $\boldsymbol{u}$ is not confined to mechanical position and momentum; it may also be decomposed into various quantities such as electric charge, voltage, or the volume of fluid in a tank, depending on the domain from which these states are derived, as shown in Table 2.

### E.2    PERFORMANCE AND COMPUTATIONAL COST

In the above case, PoDiNNs use two neural networks to approximate two energy functions $H_C$ and $H_L : \mathbb{R} \to \mathbb{R}$, thereby forming a mapping that defines an ODE, $\mathbb{R}^2 \to \mathbb{R}^2, (Q, \varphi) \mapsto (\dot{Q}, \dot{\varphi}) = (I_C, V_L)$. PoDiNNs introduce the assumption that the target system can be decomposed into interacting components and use separate neural networks $\mathbb{R} \to \mathbb{R}$ to model $n$ individual components. This approach is distinct from NODEs, which directly learn the mapping $\mathbb{R}^n \to \mathbb{R}^n$ without explicit assumptions between variables. In this way, PoDiNNs can mitigate the curse of dimensionality. Moreover, while neural networks used in NODEs are universal approximators capable of learning Kirchhoff's laws and the law of action and reaction, they inherently introduce approximation errors. This means that these laws are only approximately satisfied, and over time, these errors tend to accumulate, becoming significant. In contrast, PoDiNNs explicitly encode these laws using bivector elements, independently of the component-wise characteristics. This explicit separation enhances robustness.

If the target system includes $n$ components, we need to compute $n$ neural networks, along with the bivector. This increases the computational cost to at $n$ times that of a single NODE. We view

this computational overhead as a reasonable trade-off for the improved accuracy and interpretability provided by PoDiNNs.

## F  DATASETS AND EXPERIMENTAL SETTINGS

### F.1  IMPLEMENTATION DETAILS

Following previous studies (Greydanus et al., 2019; Matsubara et al., 2020), we used fully-connected neural networks with two hidden layers to implement any vector fields and energy functions for all models. Each hidden layer had 200 units, followed by a hyperbolic tangent activation function. Weight matrices were initialized using PyTorch's default algorithm.

In PoDiNNs, each element of the bivector $B$ related to energy-dissipating components was initialized from a uniform distribution $U(-0.1, 0.1)$, while the remaining elements are set to zero. These bivector elements were updated along with the neural network parameters. Elements representing incompatible couplings are constrained to be zero. For instance, masses cannot couple directly with each other but can couple with springs, dampers, or external forces.

Unless stated otherwise, the time step size was set to $\Delta t = 0.1$. Each training subset consisted of 1,000 trajectories of 1,000 steps, and each test subset consisted of 10 trajectories of 10,000 steps. The Dormand–Prince method (dopri5) with absolute tolerance $\text{atol} = 10^{-7}$ and relative tolerance $\text{rtol} = 10^{-9}$ was used to integrate the ground truth ODEs and neural network models (Dormand & Prince, 1986). The Adam optimization algorithm (Kingma & Ba, 2015) was applied with parameters $(\beta_1, \beta_2) = (0.9, 0.999)$ and a batch size of 100 for updates. The learning rate was initialized at $10^{-3}$ and decayed to zero using cosine annealing (Loshchilov & Hutter, 2017). The number of training iterations was set to 100,000.

All experiments were conducted on a single NVIDIA A100 GPU.

### F.2  MECHANICAL SYSTEMS

**Overview and Experimental Setting**  A spring $k$ generates a force that restores it to its original length when stretched or compressed. Specifically, this force is given by $e^S = -k(\Delta q)$, where $\Delta q$ is the spring's displacement. In both the Hamiltonian and our formulations, the potential energy $U$ is obtained by integrating this force $e^S$ with respect to the displacement $\Delta q$. Conversely, the force $e^S$ is obtained as to the partial derivative of the potential energy $U$ with respect to the displacement $\Delta q$. The spring's flow $f^S$ is the rate of extension, $\Delta \dot{q}$.

A mass $m$ has a velocity $v$, and its momentum is given by $p = mv$. The kinetic energy is $\frac{1}{2}mv^2 = \frac{1}{2m}p^2$. In general, velocity $v$ is more easily observed than momentum $p$ as the state of a mass. Therefore, we provided velocity $v$ as the observation.

Coupling between two components means the output (effort) of one flows into the input (flow) of the other. Hence, possible couplings are limited to interactions between potential and kinetic components, as shown in Table 2. Specifically, the following couplings are possible: mass and spring $\frac{\partial}{\partial p_i} \wedge \frac{\partial}{\partial q_j}$, mass and damper $\frac{\partial}{\partial p_i} \wedge \xi_j^R$, mass and external force $\frac{\partial}{\partial p_i} \wedge \xi_j^I$, moving wall and spring $\xi_i^I \wedge \frac{\partial}{\partial q_j}$, and moving wall and damper $\xi_i^I \wedge \xi_j^R$. These combinations were incorporated as elements of the learnable bivector $B$, while all other possible pairings were fixed to zero.

In the absolute coordinate system, the positions $q_i$ are used to represent the states of the springs, and their displacements $\Delta q_i$ are then calculated within the potential energy function $U$. Therefore, the potential energy function $U$ depends on all positions $q_i$ of springs and moving boundaries collectively. In the relative coordinate system, the displacements $\Delta q_i$ are used instead, and in this case, the potential energy function $U_i$ is defined individually for each displacement $\Delta q_i$.

In both Dissipative SymODENs and PoDiNNs, since we assume to know the symbolic expression $\frac{1}{2m}p^2$ for the kinetic energy, this was used, rather than a neural network. Also, the velocity $v$ was transformed into momentum $p$ using the learnable parameter $m$ with $p = mv$. This approach was originally adopted by Dissipative SymODENs and is a realistic and practical choice.

For Dissipative SymODENs and PoDiNNs in the absolute coordinate system, a single neural network was trained to approximate the potential energy function $U$ using all positions $q_i$ of springs and moving boundaries as inputs. In the relative coordinate system, a separate neural network $U_i$ was used for the potential energy of each spring $k_i$.

For Dissipative SymODENs, the dissipative terms $D$ and input gain $G$ were modeled using neural networks. In the original paper (Zhong et al., 2020), $D$ was defined as a symmetric matrix that depended solely on the positions $q$ of the springs, which is suitable for modeling linear dampers. To extend this to nonlinear dampers, we modified $D$ to also depend on the velocity $v$ of the masses.

**(a) Mass-Spring-Damper System with External Force**  This system consists of three springs $k_i$ and three masses $m_i$ arranged sequentially and indexed by $i \in \{1, 2, 3\}$ from a fixed wall. Two dampers $d_1$ and $d_3$ are placed in parallel with springs $k_1$ and $k_3$, respectively. An external force $F$ is applied to mass $m_3$.

The masses were set to $m_i = 0.8 + 0.2i$. The characteristics of the nonlinear spring were given by $k_i(\Delta q_i) = (0.1 + 0.1i)\Delta q_i + 0.1\Delta q_i^3$. The characteristics of the nonlinear dampers were defined as $d_i(v_i) = (0.1 - 0.02i)\,\mathrm{sgn}(v_i)|v_i|^{1/3}$, where $v_i$ denotes the extension velocity of the $i$-th damper, and $\mathrm{sgn}$ is the sign function that returns 1 for a positive value and $-1$ for a negative value. The initial positions of the springs and the initial velocities of the masses were sampled from the uniform distributions $U(-0.5, 0.5)$ and $U(-0.3, 0.3)$, respectively. The external force $F(t) = e^I(t)$ was defined as the sum of three sine waves, with each wave's amplitude, angular velocity, and initial phase sampled from the uniform distributions $U(0.2, 0.5)$, $U(0.1\pi, 0.2\pi)$, and $U(0, 2\pi)$, respectively.

**(b) Mass-Spring-Damper System with Moving Boundary**  This system consists of three springs $k_i$ and three masses $m_i$ arranged sequentially and indexed by $i \in \{1, 2, 3\}$ from a moving wall $b$. Three dampers $d_i$ are placed in parallel with springs $k_i$. This potentially represents a building's response during an earthquake.

The masses were set to $m_i = 1.6 - 0.2i$. The characteristics of nonlinear spring were given by $k_i = (0.6 - 0.1i)\Delta q_i + 0.1\Delta q_i^3$, where $\Delta q_i$ denotes the displacement of the $i$-th spring. The characteristics of nonlinear damper were defined as $d_i(v_i) = \tilde{d}_i \,\mathrm{sgn}(v_i)|v_i|^{1/3}$ for $\tilde{d}_1 = 0.10$, $\tilde{d}_2 = 0.05$, and $\tilde{d}_3 = 0.02$. The initial positions of springs $k_i$ and the initial velocities of masses $m_i$ were sampled from the uniform distributions $U(-0.5, 0.5)$ and $U(-0.3, 0.3)$, respectively. The position $q_b(t)$ of moving wall $b$ was set as the sum of three sine waves, with each wave's amplitude, angular velocity, and initial phase sampled from the uniform distributions $U(0.2, 0.4)$, $U(0.05\pi, 0.2\pi)$, and $U(0, 2\pi)$, respectively.

In the absolute coordinate system, it is necessary to represent the potential energy of spring $k_1$ connected to the moving wall $b$. For both Dissipative SymODENs and PoDiNNs, the position $q_b$ of the moving wall $b$ was included as part of the inputs to the neural network $U$ for the potential energy, along with the positions $q_i$ of all springs $k_i$. It was also fed to Neural ODEs.

The effort $e^I$ of the moving wall $b$ was its velocity $v_b$, which was fed to PoDiNNs and Neural ODEs as part of the external inputs. However, no such mechanism exists for Dissipative SymODENs. Without this, models cannot represent the force generated by damper $d_1$, connected to moving wall $b$.

**(c) Mass-Spring System with Redundancy**  This system consists of five springs $k_i$ and two masses $m_i$ arranged in 2-dimensional space, as shown in Fig. 2 (c). Masses $m_1$ and $m_2$ are connected to a fixed wall via springs $k_1$ and $k_2$, respectively. These masses are also connected to each other by spring $k_3$. Additionally, springs $k_4$ and $k_5$ diagonally connect masses $m_2$ and $m_1$ to the fixed wall, respectively, similar to cross braces. Since there are no energy-dissipating components or external inputs, the total energy is conserved. The masses were set to $m_1 = 5.0$ and $m_2 = 3.0$. The natural lengths of spring $k_1$ and $k_2$ to $l_1 = 3.0$, that of $k_3$ to $l_3 = 4.0$, and those of $k_4$ and $k_5$ to $l_4 = 5.0$. The characteristics of nonlinear spring were $k_1(\Delta q) = 2.5\Delta q + 3.4\Delta q^3$, $k_2(\Delta q) = 3.0\Delta q + 0.5\Delta q^3$, $k_3(\Delta q) = 2.1\Delta q + 4.1\Delta q^3$, $k_4(\Delta q) = 3.5\Delta q + 2.4\Delta q^3$, $k_5(\Delta q) = 2.5\Delta q + 1.6\Delta q^3$ for the displacement $\Delta q$. The initial positions of masses $m_1$ and $m_2$ were sampled from uniform distributions $U(-0.5, 0.5)$ in both $x$ and $y$ directions, and their velocities from $U(-0.1, 0.1)$.

Because of two masses in 2-dimensional space, the system has 4 degrees of freedom for configuration, or 8 when including velocities. However, the observations were composed of the velocities of both masses and the displacements of all five springs, resulting in a 14-dimensional observation space $\mathcal{M}$. Hence, the dynamics is degenerate.

Due to the degeneracy, HNNs are not applicable (Greydanus et al., 2019). Also, since this degeneracy does not come from a holonomic constraint of the form that CHNNs assumed, CHNNs are also not applicable (Finzi et al., 2020). In PNNs, we used a Real-NVP consisting of four coupling layers for the coordinate transformation. Each coupling layer was made of a fully-connected neural network with two hidden layers of 200 units, followed by a hyperbolic tangent activation function. The 14-dimensional observations were transformed, and the 8 dimensions were extracted (with the remaining 6 dimensions considered constant) as input to the energy function of HNNs.

## F.3 ELECTRICAL SYSTEMS

**Overview and Experimental Settings**   Refer to Appendix C.2 for basic characteristics of capacitors and inductors. Since the electric charge is analogous to displacement, a capacitor corresponds to a spring, while an inductor corresponds to a mass in mechanical systems.

The states of a capacitor $C$ and inductor $L$ are the electric charge $Q$ and magnetic flux $\varphi$, respectively, but these are difficult to observe directly. We used the capacitor voltage $V_C = Q/C$ and inductor current $I_L = \varphi/L$ as the observations, which linearly correlate with their states. PoDiNNs learned the element characteristics $C$ and $L$ as learnable parameters and internally performed the transformations $Q = CV_C$ and $\varphi = LI_L$. This is the same approach used by Dissipative SymODENs for masses. Strictly speaking, since PoDiNNs can learn coordinate transformations, they can approximate the dynamics even without such transformations.

In a mass-spring-damper system, the damper always has velocity as flow and force as effort, but the flow of a resistor can be either current or voltage. This depends on the overall coupling pattern and the formulation step, though the total number remains constant. If two resistors with different types of flow are coupled, PoDiNNs would result in a DAE, which cannot be solved explicitly. Additionally, components that share the same effort type, such capacitors and direct voltage sources, may couple. In this case, an infinite current instantaneously flows into the capacitor, and its voltage matches that of the direct voltage source. This dynamics also requires a DAE, rather than an ODE. In this study, we assumed that the system does not include couplings requiring DAEs. Methods capable of handling such circuits will be a topic for future research.

A separate neural network was used for approximating the characteristics of each energy-storing and dissipating component.

**(d) FitzHugh-Nagumo Model**   FitzHugh-Nagumo model is a model for the electrical dynamics of a biological neuron, exhibiting oscillatory behavior when the external current $J$ is applied (Izhikevich & FitzHugh, 2006). The governing equations are written as:

$$\dot{V} = V - \frac{V^3}{3} - W + J, \tag{A1}$$
$$\dot{W} = 0.08(V + 0.7 - 0.8W),$$

where $V$ denotes the membrane potential, $W$ is the recovery variable, and $J$ is the input.

A circuit representation consists of a resistor $R_2$, an inductor $L$, a capacitor $C$, a direct voltage source $E$, and a tunnel diode $R_1$, with an external current $J$. The characteristics of these elements are defined as $L = 1/0.08$, $R_2 = 0.8$, $C = 1.0$, and $E = -0.7$. The current $I_{R_1}$ through tunnel diode $R_1$ is characterized by $I_{R_1} = D(V_{R_1}) = V_{R_1}^3/3 - V_{R_1}$, where $V_{R_1}$ is the voltage across the diode. The membrane potential $V$ and recovery variable $W$ are represented by the capacitor voltage $V_C$ and the inductor current $I_L$. In our formulation, resistor $R_2$ and diode $R_1$ have the current and voltage as their flows, respectively. Also, because the voltage generated by the direct voltage source $E$ is unchanged for any trials, it can be treated as a part of resistor $R_2$.

The initial values of $V$ and $W$ were sampled from the uniform distribution $U(-3.0, 3.0)$. The external current $J$ was sampled at evenly spaced intervals within the range of 0.1 to 1.5 for each trajectory, and it was kept constant within each individual trajectory to evaluate whether each model can learn the oscillatory behavior.

Since the external current $J$ was kept constant, the variability of the dynamics was reduced, making the learning process easier. To preserve the challenge of the task, only 30 trajectories were generated for the training subset.

**(e) Chua's Circuit** Chua's circuit is a nonlinear electronic circuit known for its chaotic behavior (Chua, 2007). It consists of linear elements (a resistor $R_1$, two capacitors $C_1$ and $C_2$, and an inductor $L$) and a nonlinear element $R_2$, known as Chua's diode. With $R_1 = 1$, the governing equations are:

$$
\begin{aligned}
\dot{V}_{C_1} &= \alpha(V_{C_2} - V_{C_1} - f(V_{R_2})), \\
\dot{V}_{C_2} &= V_{C_1} - V_{C_2} + I_L, \\
\dot{I}_L &= -\beta V_{C_2}, \\
V_{R_2} &= V_{C_1}.
\end{aligned}
\tag{A2}
$$

The parameters are $\alpha = 1/C$ and $\beta = 1/L$. The nonlinear function $f(V_{R_2})$ describes the voltage-current characteristic of Chua's diode:

$$
f(V_{R_2}) = m_1 V_{R_2} + 0.5(m_0 - m_1)(|V_{R_2} + 1| - |V_{R_2} - 1|),
\tag{A3}
$$

where $V_{R_2}$ is the voltage across the diode, equal to $V_{C_1}$, and $m_0$ and $m_1$ are parameters. In our formulation, both resistor $R_1$ and diode $R_2$ have voltages as flows.

We set $\alpha = 15.6$, $\beta = 28$, $m_0 = -8/7$, and $m_1 = -5/7$. The initial values of $V_{C_1}$, $V_{C_2}$, and $I_L$ were sampled from the uniform distribution $U(-0.5, 0.5)$.

Due to its chaotic behavior, we set the time step size to $\Delta t = 0.01$, step number of test subset to 3,000 and the number of training iterations to 1,000,000.

### F.4 MULTIPHYSICS

**Overview** A multiphysics system involves components from different domains that are coupled together. Our formulation inherently handles multiphysics systems as long as careful attention is paid to which components can and cannot be coupled.

**(f) DC Motor** In this system, a DC motor bridges the electro-magnetic and the rotational domains. In the electric circuit, an inductor $L$ and a resistor $R$ represent the inductance and resistance of the motor's armature winding, respectively. Additionally, a voltage source $E$ serves as an external input. A massless pendulum rod of length $l$ is attached to the DC motor. The pendulum's angle $\theta$ is measured from the vertical position, and its angular velocity is denoted by $\omega$. With a mass $m$ at the rod's end and the gravitational acceleration $g$, the pendulum has a potential energy of $-mgl\cos\theta$ and experiences a torque of $-mgl\sin\theta$. Additionally, friction $d$ occurs at the pivot point of the pendulum.

The DC motor generates torque $\tau_{DC}$ based on the current $I_{DC}$ through the armature. We assume a linear relationship with constant $K$, such that $\tau_{DC} = KI_{DC}$. Conversely, when the motor rotates at an angular velocity $\omega$, it produces a back electromotive force given by $V_{DC} = -K\omega$. The governing equations are:

$$
\begin{aligned}
\dot{\theta} &= \omega, \\
ml^2\dot{\omega} &= -mgl\sin\theta + KI - d(\omega), \\
L\dot{I} &= -\omega K + E - R(I).
\end{aligned}
$$

We set $L = 2.5$, $m = 2.0$, $l = 1.5$, $g = 1.0$, and $K = 0.5$, respectively. Also, we set $d(\omega) = 0.02\,\mathrm{sgn}(\omega)|\omega|^{1/3}$ for friction $d$, and $R(I) = 0.05I^3$ for resistor $R$. The voltage source $E(t)$ was defined as the sum of five sine waves, where the amplitude, angular velocity, and initial phase of each wave were sampled from the uniform distributions $U(0.1, 0.3)$, $U(0.05\pi, 0.3\pi)$, and $U(0, 2\pi)$, respectively.

PoDiNNs can model this multiphysics system seamlessly. In the magnetic domain, inductor $L$ has magnetic flux $\varphi = LI$ as its state, voltage as its flow, and current as its effort. For the pendulum's motion in the rotational domain, the state is angular momentum $p = ml^2\omega$, the flow is torque, and the effort is angular velocity (see Table 2). Because the properties of the armature are represented by

inductor $L$ and resistor $R$, the remaining function of the DC motor can be represented by a coupling between inductor $L$ and the pendulum's motion $p$, that is, the bivector element $K \frac{\partial}{\partial \varphi} \wedge \frac{\partial}{\partial p}$. Both inductor $L$ and mass $m$ are considered to store kinetic energies, and this type of coupling is referred to as a gyrator.

Therefore, the only notable point for the implementation is that magnetic flux $\varphi$ and angular momentum $p$, which belong to different domains, can be coupled. Since there are only a few elements in each domain, the coupling patterns are unique, and their strengths were set to 1.0 while keeping the coefficient $K$ of the bivector element $\frac{\partial}{\partial \varphi} \wedge \frac{\partial}{\partial p}$ learnable.

**(g) Hydraulic Tank**   Consider a hydraulic tank storing incompressible fluid, which belongs to the hydraulic domain and can couple with components in the mechanical domain. Let $V$ denote the volume of the fluid inside the tank. The tank has a cross-sectional area $A$ and a fluid height $h$, giving the relationship $V = Ah$. Let $\rho$ represent the density of the fluid per unit volume, and $g$ the gravitational acceleration. The pressure $p$ exerted on the bottom of the tank per unit area is given by $p = \rho g h = \frac{\rho g V}{A}$, and the total force acting on the bottom surface is $\rho g h A$.

Assuming that fluid is supplied from the bottom of the tank, the potential energy stored in the tank can be expressed as:

$$U_V = \int_0^h \rho g h A \, \mathrm{d}h = \frac{1}{2} \rho g h^2 A = \frac{\rho g}{2A} V^2.$$

Then, $\frac{\partial U_V}{\partial V} = \rho g V / A = p$.

Consider two cylinders attached in opposite directions at the bottom of the tank, with cross-sectional areas $a_1$ and $a_2$. Each cylinder contains a piston, with masses $m_1$ and $m_2$, respectively. The forces acting on the pistons are $-pa_1$ and $pa_2$, where the positive direction is towards $m_2$. When piston $m_1$ moves by a displacement $q_1$, a volume of fluid $a_1 q_1$ flows into the tank. Similarly, when piston $m_2$ moves by $q_2$, a volume $a_2 q_2$ flows out of the tank. For simplicity, we assume that these inflows and outflows occur adiabatically, without any resistance. However, compressible fluids, non-adiabatic process, fluid momentum, and fluid resistance could also be incorporated (Duindam et al., 2009).

Each piston $m_i$ moves with a velocity $v_i$, and is connected to a fixed wall via a spring $k_i$ and a damper $d_i$ for each $i$. An external force $F = e^I$ is applied to piston $m_2$.

The equations of motion for the system can be written as:

$$\dot{V} = a_1 v_1 - a_2 v_2,$$
$$m_1 \dot{v}_1 = -p a_1 - k(q_1) - d_1(v_1),$$
$$m_2 \dot{v}_2 = p a_2 - k(q_2) - d_2(v_2) + F,$$

Each piston has kinetic energy, and each spring has potential energy. The state of each piston can be expressed as momentum, $p_i = m_i v_i$, and the state of the spring can be described by the displacement $q_i$ for each $i$. Thus, the total energy of the system is given by:

$$H = \sum_{i=1}^{2} \frac{p_i^2}{2m_i} + \sum_{i=1}^{2} U_i(q_i) + \frac{\rho g}{2A} V^2,$$

where $U_i$ denotes the potential energy stored in spring $k_i$, as well as the characteristics of dampers $d_i$.

The above equations of motion can be expressed using the energy function $H$ with the following bivector $B$:

$$
\begin{aligned}
B =& a_1 \left( \frac{\partial}{\partial p_1} \wedge \frac{\partial}{\partial V} \right) - a_2 \left( \frac{\partial}{\partial p_2} \wedge \frac{\partial}{\partial V} \right) + \frac{\partial}{\partial p_1} \wedge \frac{\partial}{\partial q_1} + \frac{\partial}{\partial p_2} \wedge \frac{\partial}{\partial q_2} \\
& + \frac{\partial}{\partial p_1} \wedge \xi_1^R + \frac{\partial}{\partial p_2} \wedge \xi_2^R - \frac{\partial}{\partial p_2} \wedge \xi^I,
\end{aligned}
$$

where $\xi_i^R$ and $\xi^I$ represent the basis vectors of the spaces $\mathcal{F}^R$ and $\mathcal{F}^I$ for dampers $d_i$ and external force $F$, respectively.

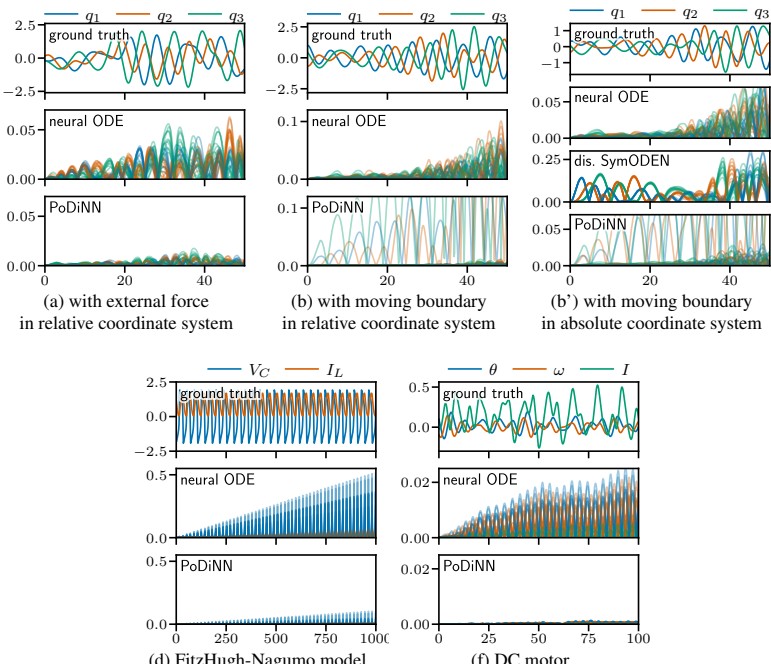

Figure A1: Visualizations of example results. Each top panel shows ground truth trajectories, while the other panels show the absolute errors of all 10 trials in semi-transparent color. See also Fig. 3.

From this, it seems that the hydraulic tank behaves similarly to a spring, but the key difference is that the coupling strength depends on the cross-sectional areas $a_1$ and $a_2$ of the cylinders. This type of coupling is referred to as a transformer. If the cross-sectional area of the tank is not constant, the tank would behave like a nonlinear spring.

We set the parameters as follows: $A = 5.0$, $g = 1.0$, $\rho = 10.0$, $a_1 = 1.0$, $a_2 = 0.3$, $m_1 = 3.0$, and $m_2 = 1.0$. The spring forces were defined as $k_i(q_i) = 0.1q_i + 0.01q_i^3$, and the damping forces as $d_i(v_i) = (0.1 - 0.04i)\,\text{sgn}(v_i)|v_i|^{1/3}$ for $i = 1, 2$.

As the system tends to reach equilibrium around $V = 5$, $q_1 = -10$, and $q_2 = 6$, the initial fluid volume $V$ and the springs' displacements $q_1$ and $q_2$ were sampled from the uniform distributions $U(5 - 0.25, 5 + 0.25)$, $U(-10 - 0.3, -10 + 0.3)$, and $U(6 - 0.3, 6 + 0.3)$, respectively, The initial velocities of the pistons were sampled from the uniform distribution $U(-0.3, 0.3)$. The external force $F(t) = e^I(t)$ was defined as the sum of three sine waves, where the amplitude, angular velocity, and initial phase of each wave were sampled from the uniform distributions $U(0.05, 0.2)$, $U(0.1\pi, 0.3\pi)$, and $U(0, 2\pi)$, respectively.

In Fig. 3, for clarity, we displayed the changes relative to $5$, $-10$, and $6$, rather than the actual values of $V$, $q_1$, and $q_2$.

## G  Additional Results and Discussions

### G.1  Additional Results

**Additional Visualization**   We show example visualizations of trajectories and absolute errors in Fig. A1 in addition to Fig. 3. Across all datasets, PoDiNNs demonstrate superior predictive performance compared to other methods. However, in both systems (b) and (b'), there was one failure. This was due to minor inaccuracies in the learned bivectors, as discussed later. While this issue only occurred twice out of 90 trials, it indicates that there is room for further improvement of the initialization and regularization of bivector elements.

**Identifying Coupling Patterns of System (b)**  We examined how the bivector $B$ identifies the coupling patterns using system (b), that is, the mass-spring-damper system with moving boundary. When two coefficients differ by a factor of 1,000 or more, we considered the larger one as a detected coupling and the smaller one as effectively zero, indicating no coupling. The basis vector $\frac{\partial}{\partial q_i}$ corresponds to spring $k_i$, $\frac{\partial}{\partial p_i}$ to mass $m_i$, and $\xi^S$ to moving boundary $b$.

In the relative coordinate system, we obtained the bivector $B = (\frac{\partial}{\partial p_1} - \xi^S) \wedge (\frac{\partial}{\partial q_1} + a_1 \xi_1^R) + (\frac{\partial}{\partial p_2} - \frac{\partial}{\partial p_1}) \wedge (\frac{\partial}{\partial q_2} + a_2 \xi_2^R) + (\frac{\partial}{\partial p_3} - \frac{\partial}{\partial p_2}) \wedge (\frac{\partial}{\partial q_3} + a_3 \xi_3^R)$ in all 10 trials. Here, the basis vectors $\xi_1^R, \xi_2^R$, and $\xi_3^R$ for dampers $d_1$, $d_2$, and $d_3$ were appropriately reordered because their indices are interchangeable. $a_1$, $a_2$, and $a_3$ are trial-wise positive parameters. This result matches the ground truth coupling pattern in Fig. 2 (b). Notably, the coefficients of other bivector elements, such as $\xi^S \wedge \frac{\partial}{\partial q_3}$ and $\frac{\partial}{\partial p_3} \wedge \frac{\partial}{\partial q_2}$, were effectively zero, indicating that PoDiNNs correctly identified the absence of non-existent couplings.

The coefficients expected to be 1 or $-1$ were accurate to five significant figures in 9 out of 10 trials. However, in one trial, there was a small error of approximately 0.002. This trial corresponded with the failure case shown in Fig. A1. This suggests that even minor errors in the coefficients of the bivector elements can lead to significant prediction failures. Nonetheless, as mentioned earlier, such occurrences are extremely rare.

**Identifying Coupling Patterns of System (b')**  In the absolute coordinate system, we obtained the bivector $B = \frac{\partial}{\partial p_1} \wedge \frac{\partial}{\partial q_1} + \frac{\partial}{\partial p_2} \wedge \frac{\partial}{\partial q_2} + \frac{\partial}{\partial p_3} \wedge \frac{\partial}{\partial q_3} + a_1 (\frac{\partial}{\partial p_1} - \xi^S) \wedge \xi_1^R + a_2 (\frac{\partial}{\partial p_2} - \frac{\partial}{\partial p_1}) \wedge \xi_2^R + a_3 (\frac{\partial}{\partial p_3} - \frac{\partial}{\partial p_2}) \wedge \xi_3^R$ in 9 out of 10 trials, where coefficients with a difference of less than 1% were considered identical. The relationship between the masses and the springs is represented by the standard Poisson bivector $\sum_i \frac{\partial}{\partial p_i} \wedge \frac{\partial}{\partial q_i}$, as indicated by the first three terms. Due to this, PoDiNNs cannot identify their specific coupling pattern. The springs' displacements are computed internally within the potential energy function. However, we can still identify how the dampers are coupled with the masses and the moving boundary, as indicated by the latter three terms. In one trial, the coefficients of bivector elements between the dampers and masses were disorganized, and this trial corresponded with the failure case shown in Fig. A1.

**Identifying Coupling Patterns of System (c)**  We also examined case of system (c). We denote the displacement of $i$-th spring in $x$-direction by $q_{xi}$. The learned bivector $B$ was $B = \frac{\partial}{\partial m_{x1}} \wedge \frac{\partial}{\partial q_{x1}} + \frac{\partial}{\partial m_{y1}} \wedge \frac{\partial}{\partial q_{y1}} + \frac{\partial}{\partial m_{x2}} \wedge \frac{\partial}{\partial q_{x2}} + \frac{\partial}{\partial m_{y2}} \wedge \frac{\partial}{\partial q_{y2}} - \frac{\partial}{\partial m_{x1}} \wedge \frac{\partial}{\partial q_{x3}} - \frac{\partial}{\partial m_{y1}} \wedge \frac{\partial}{\partial q_{y3}} + \frac{\partial}{\partial m_{x2}} \wedge \frac{\partial}{\partial q_{x3}} + \frac{\partial}{\partial m_{y2}} \wedge \frac{\partial}{\partial q_{y3}} + \frac{\partial}{\partial m_{x2}} \wedge \frac{\partial}{\partial q_{x4}} + \frac{\partial}{\partial m_{y2}} \wedge \frac{\partial}{\partial q_{y4}} + \frac{\partial}{\partial m_{x1}} \wedge \frac{\partial}{\partial q_{x5}} + \frac{\partial}{\partial m_{y1}} \wedge \frac{\partial}{\partial q_{y5}}$ in all 10 trials. The coefficients were exactly 1.0 or -1.0, accurate to five significant figures. This is because the velocities of the masses and the displacements of the springs are all observable, and their scales are known. However, the scales of spring constants $k_i$ and masses $m_i$ still cancel each other out, which means the parameters cannot be uniquely determined for the system as a whole.

The masses and springs are coupled in either the $x$- or the $y$-directions, which implies that the data is neatly separated along the $x$- and $y$-axes. From the bivector $B$, we can say the following; mass $m_1$ is coupled with springs $k_1$ and $k_5$, while mass $m_2$ is coupled with springs $k_2$ and $k_4$; and spring $k_3$ is coupled with both masses $m_1$ and $m_2$, but the coupling to $m_1$ is in the opposite direction. This means that the coupling pattern shown in Fig. 2 (c) was fully identified. By reorganizing the expression for $B$, we get $B = \frac{\partial}{\partial m_{x1}} \wedge (\frac{\partial}{\partial q_{x1}} - \frac{\partial}{\partial q_{x3}} + \frac{\partial}{\partial q_{x5}}) + \frac{\partial}{\partial m_{y1}} \wedge (\frac{\partial}{\partial q_{y1}} - \frac{\partial}{\partial q_{y3}} + \frac{\partial}{\partial q_{y5}}) + \frac{\partial}{\partial m_{x2}} \wedge (\frac{\partial}{\partial q_{x2}} + \frac{\partial}{\partial q_{x3}} + \frac{\partial}{\partial q_{x4}}) + \frac{\partial}{\partial m_{y2}} \wedge (\frac{\partial}{\partial q_{y2}} + \frac{\partial}{\partial q_{y3}} + \frac{\partial}{\partial q_{y4}})$, which indicates that system (c) has eight degrees of freedom. In this way, PoDiNNs can identify the constraints and handle degenerate dynamics.

**Impact of Number of Hidden Components for System (d)**  In the electric circuits, the flow for resistors and diodes can be either current or voltage, depending on their coupling with other components. Let $n_d$ denote the assumed number of resistors whose flow is current, and $n_g$ the assumed number of resistors whose flow is voltage. We tested system (d), the FitzHugh-Nagumo model, to explore the impact of the assumed numbers $n_d$ and $n_g$, and summarized the results in Table A1. The correct numbers for this model are $n_d = 1$ and $n_g = 1$. When fewer components are assumed, VPT values became significantly poor. On the other hand, assuming more components yields similar accuracy to the correct configuration. This trend is consistent in both the training and

test subsets. Therefore, similar to the case with system (b) in Table 4, we can identify the number of unobservable energy-dissipating components and their flow types using the training subset.

### G.2 ADDITIONAL DISCUSSIONS

**Non-Identifiability** The characteristic scales of the energy-storing and dissipating components (e.g., spring constants $k$, masses $m$, and damper constants $d$) and the coefficients of the bivector elements among them cancel each other out. Hence, the overall scale of the system is indeterminate. Once the coupling pattern has been identified, the coefficients of the bivector elements can be normalized, as shown in Section 4.2. If external forces are present, they determine the scale of force, which, in turn, uniquely determines the scale of the coupled mass. This cascading effect determines the system's overall scale.

When the characteristics of elements are linear, different coupling patterns can result in the same dynamics. Additionally, a weighted average of these coupling patterns can also be a valid solution.

While non-identifiability may be problematic for system identification, it does not impact prediction accuracy.

Table A1: Impact of # Components and VPT.

| PoDiNNs | Training | Test |
|---|---|---|
| $n_d = 0, n_g = 0$ | $0.000 \pm 0.000$ | $0.000 \pm 0.000$ |
| $n_d = 1, n_g = 0$ | $0.000 \pm 0.000$ | $0.000 \pm 0.000$ |
| $n_d = 2, n_g = 0$ | $0.000 \pm 0.000$ | $0.000 \pm 0.000$ |
| $n_d = 3, n_g = 0$ | $0.000 \pm 0.000$ | $0.000 \pm 0.000$ |
| $n_d = 4, n_g = 0$ | $0.000 \pm 0.000$ | $0.000 \pm 0.000$ |
| $n_d = 5, n_g = 0$ | $0.000 \pm 0.000$ | $0.000 \pm 0.000$ |
| $n_d = 0, n_g = 1$ | $0.007 \pm 0.000$ | $0.000 \pm 0.000$ |
| $n_d = 0, n_g = 2$ | $0.007 \pm 0.000$ | $0.000 \pm 0.000$ |
| $n_d = 0, n_g = 3$ | $0.007 \pm 0.000$ | $0.000 \pm 0.000$ |
| $n_d = 0, n_g = 4$ | $0.007 \pm 0.000$ | $0.000 \pm 0.000$ |
| $n_d = 0, n_g = 5$ | $0.007 \pm 0.000$ | $0.000 \pm 0.000$ |
| $n_d = 1, n_g = 1$ | $\mathbf{0.985} \pm 0.002$ | $\mathbf{0.640} \pm 0.070$ |
| $n_d = 1, n_g = 2$ | $\mathbf{0.985} \pm 0.000$ | $\mathbf{0.630} \pm 0.038$ |
| $n_d = 2, n_g = 1$ | $\mathbf{0.985} \pm 0.003$ | $\mathbf{0.681} \pm 0.062$ |
| $n_d = 2, n_g = 2$ | $\mathbf{0.985} \pm 0.006$ | $\mathbf{0.656} \pm 0.108$ |
| $n_d = 2, n_g = 3$ | $\mathbf{0.984} \pm 0.001$ | $\mathbf{0.577} \pm 0.072$ |
| $n_d = 3, n_g = 2$ | $\mathbf{0.978} \pm 0.008$ | $\mathbf{0.553} \pm 0.061$ |
| $n_d = 3, n_g = 3$ | $\mathbf{0.984} \pm 0.008$ | $\mathbf{0.557} \pm 0.040$ |
| | $\theta = 10^{-3}$ | $\theta = 10^{-3}$ |

**Passivity and Convexity** In understanding the behavior of dynamical systems, passivity can be a crucial concept (Khalil, 2002). Passivity implies that the system always dissipates energy and, without external inputs, converges to a stable invariant set. In PoDiNNs, general neural networks are used to model energy-dissipating components, which may sometimes learn energy-supplying characteristics, like a negative resistance. This flexibility is beneficial for learning special diodes used in the FitzHugh-Nagumo model or Chua's circuit, but it poses a problem when passivity is desired in the system. A simple solution to this issue is to replace the general neural networks with ones specialized for functions that have zero output for zero input and are monotonic non-decreasing (Wehenkel & Louppe, 2019). This ensures the system remains passive. If convexity is required in the energy function, a neural network enforcing this property can be introduced (Amos et al., 2017).

PoDiNNs decompose coupled systems into individual components, along with their coupling patterns, and learn the specific characteristics of each component. This structure allows for the introduction of constraints beforehand and ensure each component to meet the necessary properties. This flexibility is another key advantage of PoDiNNs.

