# OpenReview forum: "Poisson-Dirac Neural Networks for Modeling Coupled Dynamical Systems across Domains"
_ICLR.cc/2025/Conference — ICLR 2025 Poster_

### Official Review · Reviewer_pxB8 · 2024-11-01

**Soundness:** 3
**Presentation:** 2
**Contribution:** 3
**Rating:** 6
**Confidence:** 2

**Summary:**

This paper introduces a novel neural network architecture, the Poisson-Dirac Neural Network (PDNN), which combines Hamiltonian and Poisson formulations. The PDNN leverages the Dirac structure to model coupled dynamical systems across various domains.

**Strengths:**

1) The paper provides a solid theoretical foundation for the proposed PDNN architecture.

2) The authors conduct extensive experiments to demonstrate the effectiveness of the PDNN in various applications.

**Weaknesses:**

1) While the paper's theoretical contributions are significant, additional background material, such as differential geometry, etc, in the supplementary information would enhance its accessibility to a broader audience.

**Questions:**

1) While the PDNN shows promising results, it would be interesting to compare its performance with state-space model (SSM) based neural operators on the datasets used in this paper.

2) To make the paper more accessible to a broader audience, the authors could consider adding an appendix to provide additional explanations and visualizations of the key concepts and techniques.

---

> ### Author Response · Authors · 2024-11-18
>
> Thank you for your thoughtful feedback and constructive suggestions. We appreciate the recognition of our paper's solid theoretical foundation and extensive experiments to demonstrate the effectiveness. Below, we address each of the points raised.
>
> > W1: enhance its accessibility to a broader audience
>
> Thank you for this valuable suggestion. To the best of our knowledge, there is no comprehensive textbook specifically dedicated to analytical mechanics using Dirac structures. For an accessible introduction to differential geometry, we will newly cite Abraham et al., Manifolds, Tensor Analysis, and Applications, Springer, 1988. Regarding the relationship between differential geometry and analytical mechanics, we will refer readers to the already-cited Marsden & Ratiu (1999). The port-Hamiltonian formulation, which closely relates to our Poisson-Dirac framework, is thoroughly documented in van der Schaft & Jeltsema (2014). Finally, for a rigorous treatment of the Poisson-Dirac formulation itself, Courant (1990) is a key reference.
> We will ensure that these references, along with their explanations, are included at the beginning of Section 3 to better guide readers.
>
> > Q1: it would be interesting to compare its performance with state-space model (SSM) based neural operators on the datasets used in this paper.
>
> To our best knowledge, neural operators are primarily designed to learn mappings from functions to functions, and hence they are particularly suited for PDEs. In contrast, our work focuses on learning ODEs as mappings from vectors to vectors. Thus, the objectives differ significantly.
> Additionally, SSMs generally incorporate hidden states, whereas our study assumes that all states are directly observable. This further differentiates the objectives of the two approaches.
> If one were to construct SSMs without neural operators or hidden states, they would effectively reduce to a variant of NODEs. Therefore, a direct comparison with such an approach would not provide meaningful insights.
>
> That said, we agree that extending PoDiNNs to handle PDEs and hidden states is an exciting and important avenue for future work. Combining PoDiNNs with neural operators could be particularly promising in such contexts. We will elaborate on this direction in the final version of the paper.
>
> > Q2: the authors could consider adding an appendix to provide additional explanations and visualizations of the key concepts and techniques.
>
> Thank you for your guidance. We have updated the submitted PDF to replace Fig. 1 and add a new subsection at the end of the manuscript.
> In the updated Fig. 1, we have clarified that each arrow represents a corresponding flow or effort variable and that the Dirac structure $D$ couples these variables. Additionally, the newly added subsection provides a matrix-based representation of PoDiNNs and demonstrates how PoDiNNs reduce to HNNs under specific conditions. We believe this will help clarify how PoDiNNs extend the basic HNNs and make them more accessible to a broader audience.

---

> > ### Comment · Reviewer_pxB8 · 2024-11-25
> >
> > I want to thank the author for the rebuttal. Going through all the reviewer and rebuttal responses, I want to keep my score.

---

### Official Review · Reviewer_F183 · 2024-11-01

**Soundness:** 3
**Presentation:** 2
**Contribution:** 4
**Rating:** 8
**Confidence:** 2

**Summary:**

This paper introduces a new method for stable deep-learning based modelling of dynamical systems. Neural ODEs were originally found to have significant limitations in modelling dynamics long-term due to a lack of physical guarantees (e.g. conservation laws) leading to error accumulation. To address this, Hamiltonian NNs incorporate Hamiltonian mechanics (modelling the hamiltonian through a NN). This paper contributes a generalization of Hamiltonian NNs; Poinsson-Dirac Neural Networks (PoDiNNs) to combine both port-hamiltonian and poisson systems for mechanics, enabling learning of coupled dynamical systems across multiple domains, their interactions and degeneracies. This  constitutes a unification of previous works that address/leverage specific properties of dynamical systems like energy dissipation and external inputs. Authors show that this approach moreover allows for the identification of the specific coupling patterns between interacting dynamical systems. The experimental results show clear and consistent improvements over previous approaches in a range of systems with degenerate dynamics (e.g. constraints), dissipation and external inputs.

**Strengths:**

- The paper is well-written and concise. The modelling choices made by the authors are well-motivated; the incorporation of Dirac structure into a DL-based solving method for mechanics has a lot of possible benefits regarding guarantees and interpretability of the learned functions.
- The experimental results are very convincing; in the chosen experimental setups the proposed framework outperforms all baseline methods consistently.
- The experiments on identifying coupling patterns are fascinating, these results show that the proposed framework is indeed able to recover the underlying coupling in a very interpretable manner.

**Weaknesses:**

- The paper is quite dense, making some of the reasoning and motivation hard to follow. Although in places authors elaborate on their arguments with examples, I think it would be good to provide some more visual guidance. For example, figure 1 contains a very high-level overview of the concept of PoDiNNs, but I think it would be good to expand this diagram and clearly indicate what parts of this system the method proposes to replace with learned functions (NNs) and relate it to e.g. eq 5, def 3. I think this would greatly help with readability.
- The experiments chosen by the authors are examples of systems that fit the modelling criteria set by the authors for their framework. In these settings, the proposed framework outperforms previous approaches. However, I think it would also be good to consider/experiment with settings that do not exhibit e.g. degeneracy just to gauge how well this generalized method works in the settings that the originally proposed HNN and LNN are validated on. Would it for example be possible to show performance of your model vs baselines on eg double pendulum (as in HNN or LNN paper)?
- For completeness I think it is important to compare your method against the baselines also in terms of computational complexity / overhead. Currently, no details on the computational complexity or time complexity on either training or inference of the proposed method are given. Please provide these details.
- An arguable weakness of the method is the reliance on specifying problem-specific constraints, i.e. the number of components is a model hyperparameter, but as shown in the last experiment / Appx E it is important to specify this to at least the number of underlying components in the system, or performance will be significantly impacted. Overspecifying is possible of course, and does not seem to have negative impact, but induces additional computational complexity.

**Questions:**

See above.

---

> ### Author Response · Authors · 2024-11-18
>
> Thank you for the thoughtful and constructive review. We deeply appreciate your recognition of our paper as well-written and concise, and for finding the identification of coupling patterns fascinating. Given your deep understanding, we are curious about the reason why the confidence level was 2.
> Below, we address each of your points in detail, and we hope that these clarifications will help improve your confidence in our work.
>
> > W1: I think it would be good to expand this diagram and clearly indicate what parts of this system the method proposes to replace with learned functions (NNs) and relate it to e.g. eq 5, def 3. I think this would greatly help with readability.
>
> Thank you for this valuable suggestion. We have updated the submitted PDF and replaced Fig. 1.  In the new figure, we clarified that each arrow represents a part of flows and efforts, and the Dirac structure $D$ couples these variables, as in Definition 3. We believe this revision provides a clearer correspondence between the components and equations.
>
> > W2: how well this generalized method works in the settings that the originally proposed HNN and LNN are validated on.
>
> Thank you for raising this important point. In the final section of the updated PDF, we have clarified that PoDiNNs reduce to HNNs under conditions where HNNs are applicable--namely, when applied to systems in Darboux coordinates without degeneracy, energy dissipation, or external forces. Consequently, a direct comparison between PoDiNNs and HNNs is not meaningful.
>
> > W3: compare your method against the baselines also in terms of computational complexity / overhead
>
> We sincerely acknowledge that PoDiNNs come with a higher computational overhead compared to the baselines.
> HNNs and LNNs use a single neural network, and dissipative SymODENs are implemented with three neural networks for $H$, $D$, and $G$.
> In contrast, PoDiNNs require a separate neural network for each nonlinear component due to their decomposition approach, along with the added computation for the bivector.
> For example, for system (a) in the relative coordinate system, PoDiNNs used five neural networks for three springs and two dampers, leading to approximately double the computational complexity compared to dissipative SymODENs.
> Nonetheless, we view this computational overhead as a reasonable trade-off for the improved accuracy and interpretability provided by PoDiNNs.
> It is worth noting that if explicit decomposition were to be omitted and multiple components were consolidated into a single neural network, the computational cost would align with that of the comparative methods, but the modeling performance would also be comparable.
>
> > W4: the reliance on specifying problem-specific constraints, i.e. the number of components is a model hyperparameter
>
> You are correct. Assuming a larger number of components, the computational overhead does indeed increase, but there is no degradation in performance. As noted earlier, we view this trade-off as justified, given the significant gains in interpretability and modeling performance that PoDiNNs provide.

---

> > ### Comment · Reviewer_F183 · 2024-11-26
> >
> > Thank you for the detailed and thorough response. I appreciate the efforts in enhancing the paper's readability. While I encourage including a qualitative comparison of computational aspects like memory usage and runtime for the camera-ready, I acknowledge that this is not a critical limitation given the paper's scope and focus.
> >
> > Regarding the authors question about my confidence level, while I found the work stimulating, my limited background in classical mechanics required quite a bit of effort to fully engage with the material. I think my review still has value and should be taken into account, as I have experience with neural ODE-based DL literature and was able to grasp the main concepts. However, I will retain my original confidence score, as I cannot assure fine-grained comprehension of the proofs and principles utilized.
> >
> > I have updated my recommendation in light of the clarifications provided.

---

> > > ### Author Response · Authors · 2024-11-27
> > >
> > > Dear Reviewer F183,
> > >
> > > Thank you for considering our comments in detail.
> > >
> > > We are encouraged by your recognition of the value of our work from the perspective of neural ODE-based DL literature.
> > >
> > > Your suggestion regarding computational complexity has been very helpful. We will ensure that an extended version of the discussion in our previous comment is incorporated into the camera-ready version.
> > >
> > > Thank you once again for your thorough review and for your valuable contributions to improving our work.
> > >
> > > Authors.

---

### Official Review · Reviewer_RzPg · 2024-11-01

**Soundness:** 3
**Presentation:** 3
**Contribution:** 3
**Rating:** 6
**Confidence:** 1

**Summary:**

This work studies the application of deep neural networks for data-driven physics simulation. Over the recent years, several neural network simulation models have been proposed among which Hamiltonian Neural Networks (HNNs) and Lagrangian Neural Networks. Although these models are formulated along physics principles and consequently enjoy improved stability of their simulations, two key limitations remain: (1) they focus on mechanical systems, as opposed to for example electric circuits or magnetic fields, and (2) they consider the system as a single monolithic entity. To alleviate these limitations, this work proposes Poisson-Dirac Neural Networks (PoDiNNs), which unifies the port-Hamiltonian and Poisson representations, and explicitly represents the coupling between internal and external components. The empirical evaluation shows that PoDiNNs enjoy stability over longer simulation horizons and in general achieve lower errors than Neural ODE or variants of HNNs.

Unfortunately, I am missing too much of the mathematics and physics background that is required to understand the paper, so I am unable to provide an informative review.

**Strengths:**

**S1:** The empirical results show that PoDiNNs achieve lower errors and provide stable predictions for longer than Neural ODEs and HNN variants. As such, the method seems effective for the tasks for which it was designed.

**Weaknesses:**

**W1:** The content might be difficult to digest for the audience of ICLR, since a specific mathematical and physics background is required to understand the work. In my impression, most ML researchers will lack this background, and also a substantial part of the AI4Science community might find it difficult to understand the paper.

**Questions:**

-

---

> ### Author Response · Authors · 2024-11-18
>
> Thank you for reviewing our manuscript and for recognizing that PoDiNNs achieve lower errors and provide stable predictions over longer time horizons.
>
> > W1: The content might be difficult to digest for the audience of ICLR, since a specific mathematical and physics background is required to understand the work.
>
> We aimed to emphasize the theoretical foundations and broad applicability of PoDiNNs, which led us to dedicate significant space to abstract discussions of differential geometry. However, the implementation itself is quite accessible. When appropriate local coordinates are introduced on the state space, a bivector is represented by a matrix, and PoDiNNs are implemented using matrix-vector multiplications. To aid in understanding, we have updated the submitted PDF file and presented the matrix-based representation in the final section, as well as its relation to prior works. We believe this will help clarify how PoDiNNs extend the basic Hamiltonian Neural Networks and make them more intuitive for a broader audience.

---

> > ### Comment · Reviewer_RzPg · 2024-11-25
> >
> > Thank you for adding this section. I do not change my review since I do not have the appropriate background to assess the paper.

---

### Official Review · Reviewer_zHG9 · 2024-11-03

**Soundness:** 3
**Presentation:** 4
**Contribution:** 3
**Rating:** 8
**Confidence:** 4

**Summary:**

This work targets the dynamical system identification using observation data, which is a hot topic and essential application. The key differences with respect to existing works are clearly stated: 1) identifying unknown physics rather than predefined symbolics, 2) considering the coupling behaviors in the system or interactions when the problem covers multiple domains rather than single mechanical systems.

**Strengths:**

•	As the reviewer summarizes above, key limitations are correctly identified such that the contributions of this work are clear. To the best of the reviewer’s knowledge, this work is new.

•	The theoretical analysis is solid, where the definitions and theorems clearly show how the Dirac structure encapsulates internal and external component couplings. Moreover, the corresponding examples of different dynamical systems are well-explained to differentiate the proposed method from existing methods like HNN and NODE.

•	The results on multiple systems look promising, where various experiment scenarios and evaluation metrics are comprehensive to validate PoDiNNs’ capabilities.

**Weaknesses:**

Some technical details, as well as the claimed capabilities, are unclear, which might be because the reviewer is unfamiliar with all kinds of multi-domain dynamical systems. The confusions are listed below.

•	The capability to deal with multi-physics problems is claimed several times in the paper. Specifically, Remark 1 explains the representation of inter-coupling using a bivector element (which is also an NN, right?) Remark 4 with Table 2 briefly demonstrates the scenarios to capture coupled physics in multiple domains. The reviewer would like to know how PoDiNNs represent such interactions.  Is it the same way as the traditional simulation tool, e.g., through iterative refinement of two (or more) simulations of a single domain or system?

•	The proposed work targets unknown physics/dynamics, which is quite challenging as there are no predefined physical symbolics in PINN-alike works. How to ensure the PoDiNNs capture the correct physics without causing overfitting problem or continuous good performance in extrapolation?

•	Moreover, PoDiNNs focus on behaviors that may not be captured by generic models, e.g., NODE, which seem to need intensive resources. Especially, the couplings in multi-physics usually require heavy computation in traditional simulations. The more fine-grained, the heavier. What is PoDiNNs’ capability in this aspect?

•	For the last paragraph of Sec. 3.4, an example of electric circuit is used. Could the reviewer further explain with more details: why ODEs or using NODE alone cannot capture the current flow and balanced voltage level? Subsequently, how does PoDiNN mitigate the issue of limited representation? A toy example with mathematical derivations or diagram will be helpful.

**Questions:**

Please refer to the bullet points in Weaknesses.

---

> ### Author Response · Authors · 2024-11-18
>
> Thank you for the thoughtful review of our paper. We sincerely appreciate your recognition of the novelty, clarity, and solid theoretical analysis of our work, as well as the comprehensive evaluation we conducted.
> We are grateful for your constructive feedback and are pleased to address your concerns:
>
> > W1: through iterative refinement of two (or more) simulations of a single domain or system?
>
> Multiphysics simulations with iterative refinement are referred to as "weak coupling."
> In this work, we aim to achieve "strong coupling," where all domains are computed simultaneously rather than iteratively.
> In Poisson-Dirac formulation, the bivector element represents the flow of physical quantities--such as force, velocity, current, or voltage--between components.
> Its skew-symmetric property ensures that energy inflows and outflows between domains remain balanced, with energy loss or supply occurring only within individual components.
> This constraint defines the interactions across domains, making "strong coupling" feasible.
> By leveraging this framework, PoDiNNs enable the data-driven learning of inter-domain interactions.
>
> > W2: How to ensure the PoDiNNs capture the correct physics without causing overfitting problem or continuous good performance in extrapolation?
>
> As you pointed out, unlike PINN-alike works, PoDiNNs do not rely on predefined physical symbolics. Instead, methods like Hamiltonian Neural Networks leverage the equation frameworks (e.g., Hamilton's equations, Euler-Lagrange equations, etc.), which ensure the correct physics to be captured.
>
> Furthermore, PoDiNNs introduce assumptions that are weaker than those in PINNs but stronger than those in Hamiltonian Neural Networks: the target system can be decomposed into interacting components. For example, in a coupled system with two masses and two springs in a 1D space, the full state space is four-dimensional (two positions and two velocities), which requires a mapping $R^4\to R^4$ to define the vector field. When decomposing the system into four separate components, we can represent it using four separate mappings $R\to R$. In this way, we can mitigate the curse of dimensionality and enhance generalization. We will add this explanation to clarify our strategy for achieving robust extrapolation.
>
> > W3: PoDiNNs focus on behaviors that may not be captured by generic models, e.g., NODE, which seem to need intensive resources.
>
> You are correct. For instance, if our target system includes four components, we need to compute four neural networks, along with the bivector. This increases the computational cost to at least four times that of a single NODE.
> We view this computational overhead as a reasonable trade-off for the improved accuracy and interpretability provided by PoDiNNs.
> We will include this clarification in the revised manuscript.
>
> > W4: Could the reviewer further explain with more details: why ODEs or using NODE alone cannot capture the current flow and balanced voltage level?
>
> Thank you for this insightful question. While neural networks used in NODEs are universal approximators capable of learning the current flows and balanced voltage levels, they inherently introduce approximation errors. In practice, this means that Kirchhoff's laws are only approximately satisfied, and over time, these errors tend to accumulate, becoming significant.
> In contrast, PoDiNNs explicitly encode the current flows and balanced voltage levels using bivector elements, leading to Kirchhoff's laws, independently of the component-wise properties. This explicit separation enhances robustness. We will include this explanation in the revised manuscript.

---

> > ### Comment · Reviewer_zHG9 · 2024-11-27
> >
> > Thanks for the point-to-point answers. I'd like to keep the paper rating. In addition, I have a follow-up question for the last response.
> >
> > "W4: Could the reviewer further explain with more details: why ODEs or using NODE alone cannot capture the current flow and balanced voltage level?
> >
> > Thank you for this insightful question. While neural networks used in NODEs are universal approximators capable of learning the current flows and balanced voltage levels, they inherently introduce approximation errors. In practice, this means that Kirchhoff's laws are only approximately satisfied, and over time, these errors tend to accumulate, becoming significant. In contrast, PoDiNNs explicitly encode the current flows and balanced voltage levels using bivector elements, leading to Kirchhoff's laws, independently of the component-wise properties. This explicit separation enhances robustness. We will include this explanation in the revised manuscript."
> >
> > Could the authors provide more detailed and technical explanations, rather than just descriptions, to show how PoDiNNs approximate the mapping and how to ensure the satisfaction of Kirchhoff's laws? For example, what is meant by "explicitly encode the current flows and balanced voltage levels using bivector elements"? If the answer is aligned with the technical contents of the manuscript, the authors are welcome to point it out. This may help raise the confidence score.

---

> > > ### Author Response · Authors · 2024-11-28
> > >
> > > Dear Reviewer zHG9,
> > >
> > > We deeply appreciate your thoughtful follow-up question and your continued engagement in the discussion.
> > >
> > > Let us consider a specific example from Appendix B.2: a system where a capacitor $C$ and an inductor $L$ are coupled in series. The state of the capacitor $C$ is its electric charge $Q$, while the state of the inductor $L$ is its magnetic flux $\varphi$. Their coupling is represented by the bivector $B = \frac{\partial}{\partial \varphi} \wedge \frac{\partial}{\partial Q}$.
> > > Using local coordinates $(Q, \varphi)$, we can represent $B$ as a matrix $\begin{bmatrix}
> > > 0 & 1\\\\
> > > -1 & 0
> > > \end{bmatrix}$.
> > > The Poisson-Dirac formulation states that applying this matrix to the effort $(\nabla_Q H_C, \nabla_\varphi H_L)$ yields the flows $(\dot Q, \dot \varphi)$, that is,
> > > \begin{equation*}
> > >   \begin{bmatrix}
> > >     I_C\\\\
> > >     V_L
> > >   \end{bmatrix}=
> > >   \begin{bmatrix}
> > >     \dot Q\\\\
> > >     \dot \varphi
> > >   \end{bmatrix}=
> > >   \begin{bmatrix}
> > >     0 & 1\\\\
> > >     -1 & 0
> > >   \end{bmatrix}
> > >   \begin{bmatrix}
> > >     \nabla_Q H_C\\\\
> > >     \nabla_\varphi H_L
> > >   \end{bmatrix}=
> > >   \begin{bmatrix}
> > >     0 & 1\\\\
> > >     -1 & 0
> > >   \end{bmatrix}
> > >   \begin{bmatrix}
> > >     V_C\\\\
> > >     I_L
> > >   \end{bmatrix}=
> > >   \begin{bmatrix}
> > >     I_L\\\\
> > >     -V_C
> > >   \end{bmatrix}.
> > > \end{equation*}
> > > This corresponds to the standard form of Hamilton's equations.
> > > We can see that the first line represents Kirchhoff's current law, $I_C=I_L$, and the second line represents Kirchhoff's voltage law, $V_L=-V_C$. The sign inversion follows from the skew-symmetric property of the bivector $B$.
> > >
> > > As the number of components increases, the dimensionality of the flows, efforts, and matrix also grows, and the bivector may not retain a standard form, as observed in system (e). Nonetheless, the bivector consistently represents Kirchhoff's laws, independent of the component-wise characteristics, and the Poisson-Dirac formulation ensures the satisfaction of these laws.
> > >
> > > PoDiNNs use two neural networks to approximate two energy functions $H_C$ and $H_L:R\to R$ in this case, thereby forming a mapping that defines an ODE,$R^2\to R^2, (Q,\varphi)\mapsto(\dot Q,\dot\varphi)=(I_C,V_L)$.
> > > In general, PoDiNNs use separate neural networks $R\to R$ for $n$ components. This approach is distinct from NODEs, which directly learn the mapping $R^n\to R^n$ without explicit assumptions between variables. Similarly, in mechanical systems, bivectors represent the law of action and reaction.
> > >
> > > To be honest, the learned bivector elements may sometimes include arbitrary coefficients. This is because the scales of input, output, and characteristics (e.g., capacitance, inductance, resistance) of a component may cancel out, making them not uniquely determined. However, as the scales of flows and efforts are typically known through measurements, these coefficients can be uniquely determined (as 0, 1, or -1 in most cases) by applying appropriate scaling. This approach was implemented for system (e) in Section 4.2.
> > >
> > > We hope this explanation addresses your question. Thank you once again for your insightful question, which provided us with the opportunity to turther clarify and refine the presentation of our work.

---

### Official Review · Reviewer_h9yM · 2024-11-03

**Soundness:** 3
**Presentation:** 3
**Contribution:** 3
**Rating:** 5
**Confidence:** 4

**Summary:**

The authors present a new architecture, specifically they integrate some clever ideas of Dirac structure in order to unify the port-Hamiltonian and Poisson formulations from geometric mechanics.  Both nice innovations for physics-constrained neural networks.

**Strengths:**

The integration of physics-based principles is always welcome in the dynamical systems world.  The authors have a very nice contribution to make here potentially as the integration of physics principles into neural networks is very important.

**Weaknesses:**

The models used seem to be all linear: which begs the question about simple linear model regressions such as dynamic mode decomposition, dynamic mode decomposition with control and time-delay embedded DMD which models missing/coupled physics.  These more baseline (non-NN) methods are simply not talked about or considered and I think they should be.

There are statements that are simply not true:  "two key limitations remain in modeling dynamical systems, especially those described by ordinary differential equations (ODEs). The first limitation is the narrow focus on mechanical systems."  This suggests the authors don't know the field well and it is a concern.  People are modeling all kinds of dynamical systems with ML/AI architectures well beyond mechanical systems.

Further: "The second limitation is that most methods treat the system as a single, monolithic entity."  This is also not true.  Many people are working on coupled systems where time-delay embeddings are often used to gather information for missing, coupled and unmeasured variables.  Again, it is odd that these statements exist in the paper which suggests the authors are not aware of the great body of work on model discovery and dynamical systems methods with ML/AI for systems which are not mechanical and which indeed have coupling.

**Questions:**

The models considered all seem to be linear.  Is that correct?  If so, more standard system ID or linear methods should be considered instead of all this sophisticated ML/AI architectures.

Can this generalize to nonlinear models?  Or simply be applied to nonlinear models with success?  Although you did Fitzhugh-Nagumo and  Chua's model, these are only "slightly" nonlinear and system ID models can work pretty well with those.

What have you not considered non-neural network methods like DMD... or system ID methods like mamba/S4 not been compared?

---

> ### Author Response · Authors · 2024-11-18
>
> Thank you for taking the time to review our manuscript and for recognizing our research as a very nice contribution to a very important topic of the integration of physics principles into neural networks. We appreciate your thoughtful feedback and would like to address the concerns you raised, some of which may stem from a lack of clarity in our explanations.
>
> > W1: The models used seem to be all linear:
> >
> > Q1: The models considered all seem to be linear.
> >
> > Q2: Can this generalize to nonlinear models? Or simply be applied to nonlinear models with success?
>
> *Our proposed method, PoDiNNs, is indeed nonlinear.* As illustrated in Fig. 4, our model successfully learns the nonlinear characteristics of Chua's diode, as well as other springs, dampers, and resistors. While the Dirac structure, symplectic forms, and Poisson bivectors are linear, this linearity is well-supported by theoretical foundations in analytical mechanics. Indeed, the significance of PoDiNNs lies in decomposing the coupled system into linear interactions and nonlinear component characteristics. This separation is essential for achieving both high modeling accuracy and interpretability.
>
> > W1: time-delay embedded DMD which models missing/coupled physics
>
> Our model identifies the characteristics of energy-dissipating components, such as diodes and dampers, whose inputs and outputs are unobservable. It is important to note that these components do not have states, and we did not attempt to estimate missing (unobservable) states. Instead, our model focuses on identifying the unknown relationships between observable states (of masses, springs, etc.) determined by unobservable components (like dampers and diodes). We acknowledge the importance of estimating unobservable states, and methods like time-delay embeddings could indeed be useful for that purpose. We will clarify in the revised manuscript that this topic is beyond the scope of our current work and is a promising direction for future research.
>
> > W2-W3: There are statements that are simply not true:
>
> The statements you mentioned were intended to reflect limitations specifically within the domain of Hamiltonian neural networks and similar approaches that integrate neural networks with analytical mechanics, as summarized in Table 1. We believe that, within this context, our claims are accurate.
>
> However, you are right that there are numerous alternative methods, such as DMD, Koopman operator theory, state-space models (including mamba/S4), and SINDy, which address a broader spectrum of systems, including non-mechanical and coupled systems. We will revise Section 2 of our manuscript to clarify our focus and provide a more comprehensive overview of related approaches to better situate our work within the broader landscape.
>
> > Q3: What have you not considered non-neural network methods like DMD... or system ID methods like mamba/S4 not been compared?
>
> General modeling methods like DMD or general system ID methods like mamba/S4 are indeed broadly applicable.
> However, because these methods typically do not ensure physical laws (such as laws of energy conservation and dissipation or Kirchhoff's laws).
> As confirmed in the prior studies summarized in Table 1, these general methods often face challenges in producing accurate long-term predictions and providing the physical interpretability.
> In contrast, as demonstrated in Section 4, PoDiNNs identify systems in the form of coupled components with individual characteristics (e.g., energy-storing or dissipating), which offers more interpretable and physically meaningful insights.
> This distinction arises from the difference in objectives, rather than a matter of superiority.
> Consequently, we believe it is appropriate to acknowledge these methods in the related work section, without making further direct comparisons.

---

> > ### Comment · Area_Chair_fcti · 2024-12-03
> >
> > Dear reviewer h9yM,
> >
> > Thank you for your review. Could you please read the authors' rebuttal and inform me if and how this impacts your standpoint, accompanied by a motivation?
> >
> > Kind regards,
> >
> > AC

---

### Comment · Area_Chair_fcti · 2024-11-25
**Last day for reviewers to ask questions to the authors!**

Dear reviewers,

Tomorrow (Nov 26) is the last day for asking questions to the authors. With this in mind, please read the rebuttal provided by the authors, as well as the other reviews. If you have not already done so, please explicitly acknowledge that you have read the rebuttal and reviews, provide your updated view _accompanied by a motivation_, and raise any outstanding questions for the authors.

**Timeline**: As a reminder, the review timeline is as follows:
- November 26: Last day for reviewers to ask questions to authors.
- November 27: Last day for authors to respond to reviewers.
- November 28 - December 10: Reviewer and area chair discussion phase.

Thank you again for your hard work,

Your AC

---

### Meta-Review · Area_Chair_fcti · 2024-12-17

**Metareview:**

The reviews for this submission were leaning towards acceptance of the paper. Among other things, reviewers highlighted as strengths the use of well-explained examples of different dynamical systems that allow to differentiate the proposed method from existing methods, and which demonstrated convincing results in terms of lower errors and more stable predictions over longer time horizons. The presentation of the paper received both praise and requests for improvement, with the accessibility of the paper being the most important improvement area. Several reviewers raised questions about the computational cost of the proposed method compared to baselines, which the authors answered transparently during the rebuttal. The reviewer with the most critical review requested comparisons against other methods such as system ID methods, but has unfortunately not engaged with the authors anymore after their rebuttal, despite requests to do so. Despite one other reviewer agreeing that the paper would be strengthened by a comparison to these types of methods, this reviewer maintained their recommendation that the paper is marginally above the acceptance threshold. Taking all of this into account, the recommendation is to accept this paper. The authors are strongly encouraged to include the additional clarifications that were helpful during the rebuttal in their camera ready version of the manuscript.

**Additional Comments On Reviewer Discussion:**

Overall, the reviewers who initially already leaned towards a positive recommendation have maintained this stance during the discussion period. The authors engaged with the reviewers and addressed questions about the computational cost of the proposed method and how this compares against baselines. The answers by the authors were clear, and they have agreed to include this in the camera ready version. The confidence of the already positively minded reviewer zHG9 has increased by the more detailed explanation provided by the authors using a specific example from Appendix B.2, which explains how PoDiNNs can ensure the satisfaction of Kirchhoff's laws.

Reviewer h9yM was the only reviewer with a recommendation that this paper is below the bar for acceptance. They raised questions about whether or not the proposed method could model non linear characteristics and requested comparison against other baseline methods than those presented in the paper. Given the lack of engagement of this reviewer, and the fact that the points raised by this reviewer were not enough to convince the other reviewers that the paper was below the bar of acceptance, the suggested points of improvements were considered outweighed by the positives.

---

### Decision · Program_Chairs · 2025-01-22

Accept (Poster)